palaeontology

dinosaur, ceratopsid, centrosaurine, evolution, anagenesis, Cretaceous

**Author for correspondence:**
John P. Wilson
e-mail: jackwilson1899@gmail.com

# A new, transitional centrosaurine ceratopsid from the Upper Cretaceous Two Medicine Formation of Montana and the evolution of the '*Styracosaurus*-line' dinosaurs

John P. Wilson[1], Michael J. Ryan[2,3] and David C. Evans[4,5]

[1]Varricchio Lab, Department of Earth Sciences, Montana State University, Bozeman, MT 59715, USA
[2]Department of Earth Sciences, Carleton University, 2125 Herzberg Building, 1125 Colonel By Drive, Ottawa, Ontario Canada K1S 5B6
[3]Department of Palaeobiology, Canadian Museum of Nature, PO Box 3443, Station 'D', Ottawa, Ontario, Canada K1P 6P4
[4]Department of Natural History, Royal Ontario Museum, 100 Queen's Park, Toronto, Ontario, Canada M5S 2C6
[5]Department of Ecology and Evolutionary Biology, University of Toronto, 25 Willcocks Street, Toronto, Ontario, Canada M5S 3B2

JPW, 0000-0002-4973-3139

Ceratopsids are among the most ubiquitous dinosaur taxa from the Late Cretaceous terrestrial formations of the Western Interior of North America, comprising two subfamilies, Chasmosaurinae and Centrosaurinae. The Two Medicine Formation of northwestern Montana has produced numerous remains of centrosaurine dinosaurs, which represent three taxa previously considered valid: *Rubeosaurus ovatus*, *Einiosaurus procurvicornis* and *Achelousaurus horneri*. Here, we reassess the previous referral of specimens to *Rubeousaurus ovatus* and demonstrate that this taxon is represented solely by its holotype specimen, which was first diagnosed as *Styracosaurus ovatus*. One of the specimens previously referred to '*Rubeosaurus*' *ovatus* instead represents a new eucentrosauran centrosaurine taxon diagnosed here, *Stellasaurus ancellae* gen. et sp. nov. *Stellasaurus* expresses a unique combination of eucentrosauran centrosaurine characters, including an elongate nasal horncore,

diminutive supraorbital horncores, and a parietal bearing straight, elongate P3 processes, semi-elongate P4 processes and non-elongate P5, P6 and P7 processes. Within the stratigraphic succession of Eucentrosaura, *Stellasaurus* occurs intermediate to *Styracosaurus albertensis* and *Einiosaurus*, and likewise reflects intermediate morphology. Assessed within the stratigraphic, geographical, taphonomic, ontogenetic and phylogenetic framework of Unified Frames of Reference, we fail to reject the hypothesis that *Stellasaurus ancellae* represents a transitional taxon within an anagenetic lineage of eucentrosauran centrosaurines.

# 1. Introduction

The Late Cretaceous Western Interior of North America played host to the rapid evolution and diversification of numerous clades of ornamented dinosaurs. Among these, ceratopsids are primarily characterized by their highly diverse and diagnostic cranial ornamentation, and were abundant large-bodied consumers in Laramidian ecosystems [1–4]. Between an estimated 90 and 80 Ma (phylogenies of [5,6]), Ceratopsidae diverged into Chasmosaurinae and Centrosaurinae, derived members of which often express differing trends in cranial ornamentation [1,7]. While derived chasmosaurines are typically characterized by elongate postorbital horns, small nasal horns and squared, elongate parietosquamosal frills, centrosaurines are typically characterized by diminutive supraorbital horns or variably sized bosses, larger nasal horncores and shorter, more rounded frills. The basalmost centrosaurines, such as *Diabloceratops* [8] and *Albertaceratops* [9], exhibit some plesiomorphic features of Ceratopsidae, such as long postorbital horncores (which are shared with *Zuniceratops*, the sister taxon to Ceratopsidae), which gave way to the more typical centrosaurine condition in certain more derived taxa.

Like many of the other terrestrial dinosaur-bearing formations of the Late Cretaceous Western Interior, the Campanian Two Medicine Formation of northwestern Montana has produced numerous remains of centrosaurine ceratopsids [10,11], from which three stratigraphically separated taxa were previously recognized: *Rubeosaurus ovatus* [12–14], *Einiosaurus procurvicornis* [15] and *Achelousaurus horneri* [15]. The first centrosaurine described from the Two Medicine Formation (and still considered valid, excluding *Brachyceratops*) was *Styracosaurus ovatus*, diagnosed by Gilmore [12] based on an isolated partial parietal, USNM 11869. The holotype of *Styracosaurus ovatus* is characterized by elongate, medially inclined P3 parietal processes, elongate P4 processes, a partially elongate P5 process, and lack of P1 processes. Three additional, stratigraphically successive centrosaurine taxa ('Taxon A', 'Taxon B' and 'Taxon C') were described by Horner *et al.* [11] and hypothesized to represent members of an anagenetic lineage, the direct descendants of *Styracosaurus albertensis* and the direct ancestors of *Pachyrhinosaurus*. Sampson [15] later named and diagnosed 'Taxon B' and 'Taxon C' as *Einiosaurus procurvicornis* and *Achelousaurus horneri*, respectively, and proposed an alternative cladogenetic origin of these taxa was more likely. *Einiosaurus* is diagnosed by a parietal with elongate, straight, posteriorly oriented P3 processes, a highly procurved nasal horncore, and rounded masses of bone forming the supraorbital ornamentation. *Achelousaurus* is characterized by a parietal with elongate, posterolaterally curved P3 processes and high-ridged supraorbital and nasal bosses. 'Taxon A', represented by Museum of the Rockies specimen MOR 492, recognized to be a presumable new taxon by Horner *et al.* [11], remained undescribed in detail until it was referred to *Styracosaurus ovatus* by McDonald & Horner [13]. It was inferred that the preserved diagnostic portion of the MOR 492 parietal, a single lateral bar, would have been oriented in such a way that the P3 processes were medially inclined, thus allowing referral to *S. ovatus*. However, in their phylogenetic analysis, *S. ovatus* was pulled away from its original sister taxon relationship with *Styracosaurus albertensis* and thus the genus name of *Styracosaurus ovatus* was replaced with *Rubeosaurus*.

Here, we demonstrate that MOR 492, along with subsequently described specimens referred to *Rubeosaurus* [14], is not referable to *S. ovatus* (figures 1–3), and consequently that *S. ovatus* is represented solely by its holotype specimen, USNM 11869. A revised phylogenetic analysis returns *S. ovatus* to a sister taxon relationship with *S. albertensis*, and as such, the genus name *Rubeosaurus* becomes unnecessary. MOR 492 instead represents a new taxon of centrosaurine ceratopsid, *Stellasaurus ancellae* gen. et sp. nov. described here, diagnosed by a unique combination of characters inferred to be intermediate or transitional between the stratigraphically preceding *Styracosaurus albertensis* and the stratigraphically successive *Einiosaurus*. The significance of *Stellasaurus* within Centrosaurinae and hypotheses regarding its evolutionary mode are assessed within the analytical framework of Unified Frames of Reference [4], which accounts for its stratigraphic, geographical and phylogenetic relationship to other centrosaurine taxa, as well as documents how its ontogenetic status and taphonomic preservation affect interpretations

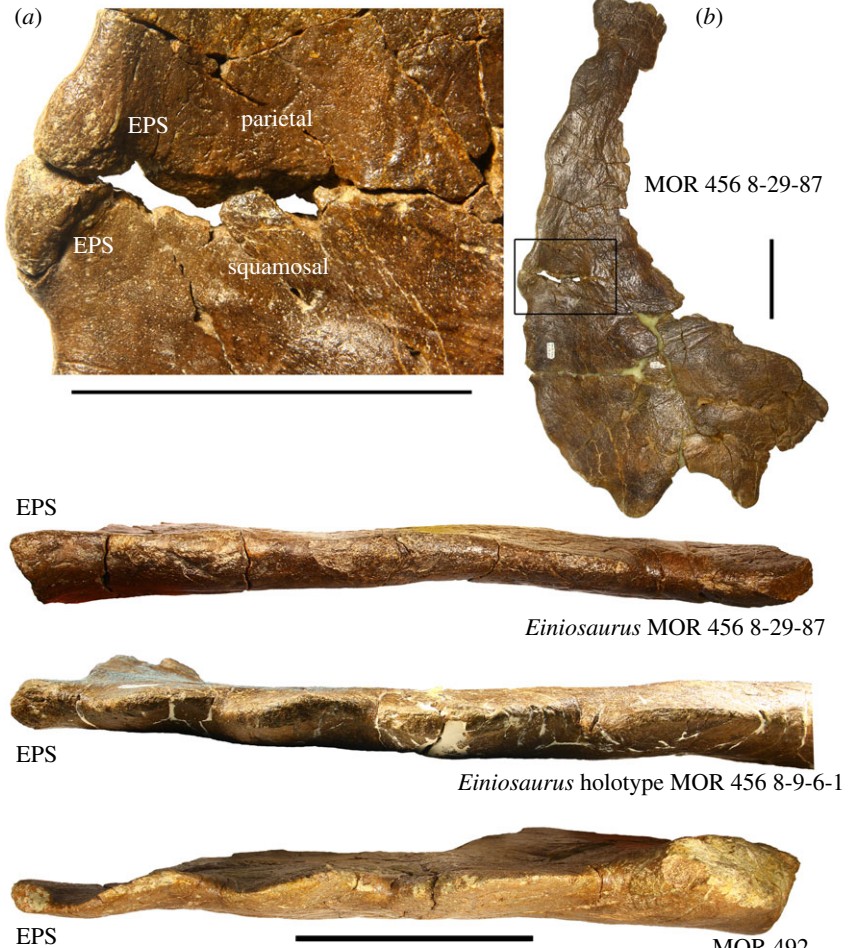

**Figure 1.** EPS structures in select centrosaurine specimens from the Upper Two Medicine Formation. EPS structures are bulbous and rounded and are dissimilar to the processes lining the margins of the parietal. Structures are labelled 'EPS' due to being analogous with true epiparietosquamosals, but are not thought to be formed from distinct epiossifications in the Two Medicine Formation centrosaurines. (*a*) Inset of (*b*) showing EPS structure divided across the parietosquamosal contact. EPS, epiparietosquamosal. Scale bars 10 cm.

of its diagnostic features. All of the available stratigraphic, geographical and corresponding transitional morphological evidence is consistent with, and fails to reject, the hypothesis that the centrosaurines of the Two Medicine Formation, along with *Styracosaurus albertensis* and possibly *Pachyrhinosaurus*, represent members of a single anagenetic lineage [16], with the evolution of their diagnostic cranial ornamentation possibly facilitated through peramorphic heterochrony. Alternative hypotheses involving cladogenetic relationships of these taxa are considered and weighed against the available evidence.

## 2. Institutional abbreviations

AMNH FARB, American Museum of Natural History, Fossil Amphibians, Reptiles, and Birds, New York, New York; CMN, Canadian Museum of Nature, Ottawa, Ontario, Canada; MOR, Museum of the Rockies, Bozeman, Montana; TMP, Royal Tyrrell Museum of Palaeontology, Drumheller, Canada; UALVP, University of Alberta Laboratory of Vertebrate Palaeontology, Edmonton, Alberta, Canada; USNM, United States National Museum of Natural History, Washington DC.

## 3. Material and methods

MOR 492, an isolated and fragmentary partial skull of a centrosaurine ceratopsid preserving the left lateral parietal bar, proximal portion of the midline parietal bar, near-complete paired and fused nasals,

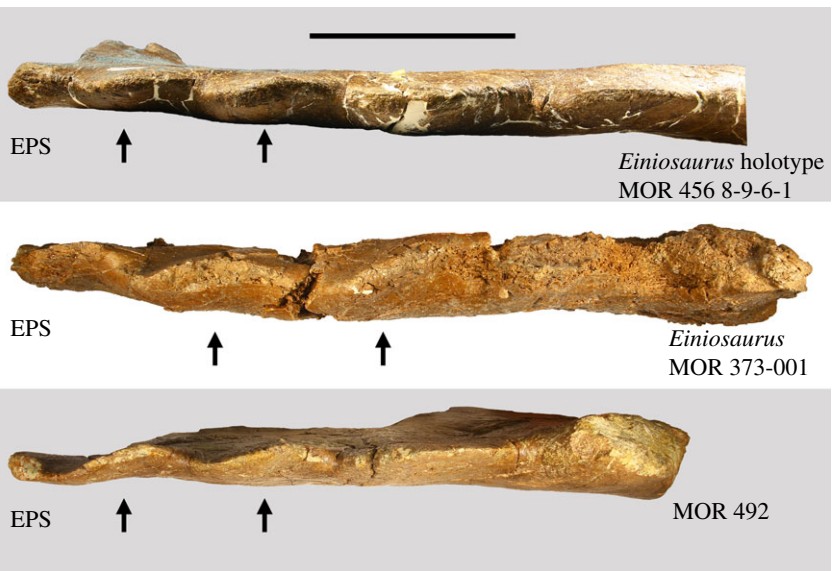

**Figure 2.** Imbrication in select centrosaurine specimens from the Upper Two Medicine Formation. Specimens are in lateral view, with anterior to left and dorsal upwards. Imbrication, indicated by black arrows, occurs in anteriormost two processes in all centrosaurines from the Two Medicine Formation. EPS, epiparietosquamosal. Scale bar 10 cm.

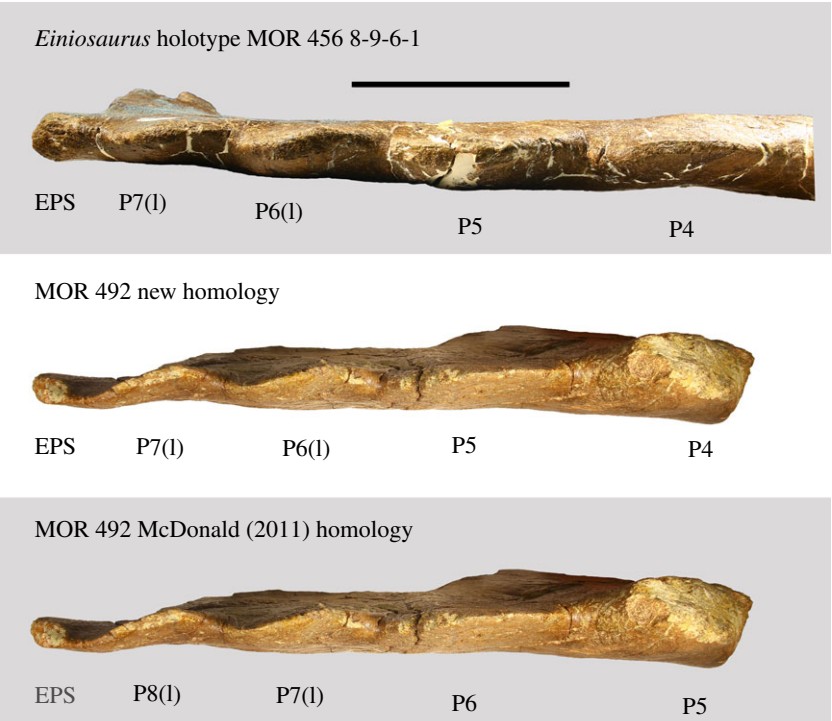

**Figure 3.** Homology of parietal processes as indicated by imbrication. EPS is greyed out in the McDonald [14] homology as it was not identified as such in that study. EPS, epiparietosquamosal; (l), imbricated. Scale bar 10 cm.

partial left premaxilla and partial left postorbital and associated supraorbital ornamentation. MOR 492 was recovered from Lithofacies 5 [17] of the uppermost Two Medicine Formation, 65 m below the upper contact between the Two Medicine Formation and the overlying Bearpaw Formation [11]. Morphological description and comparison to the type and referred specimens of *Einiosaurus* and *Achelousaurus* was conducted first hand. Correlations of geological units and corresponding biostratigraphic inferences follow the recalibrated radiometric dates of Fowler [18], which represent the most current and accurate radiometric correlations of the Late Cretaceous geological formations of the Western Interior of North America.

Both Bayesian and parsimony phylogenetic inference analyses were conducted using a modified version of the character matrix of Tykoski *et al.* [19] to assess the systematic position of *Stellasaurus ancellae*. The Bayesian analysis was conducted in MrBayes 3.2 [20] using the parameters of the Bayesian analysis conducted by Lund *et al.* [6]. Additional calibrations were added to constrain the ages for the nodes Neoceratopsia, Ceratopsidae and Centrosaurinae. Four new characters were created to account for conditions not previously included in the widely used centrosaurine character matrix. Two new characters (106, 107) were added to account for taxonomically informative varied degree of elongation of the P4 ([0] non-elongate; [1] half as long as P3 or less; [2] more than half as long to as long as P3) and P5 ([0] non-elongate; [1] half as long as P3/P4 or less; [2] more than half as long to as long as P3/P4) parietal processes. One character (108) was created to account for the variable percentage ([0] >50%–100%; [1] <50% but >0%; [2] 0%) of parietal processes which are represented by true epiossifications as opposed to outgrowths of the parietal itself; we present an alternative working hypothesis that some parietal processes of certain eucentrosaran taxa are not formed by fusion of a separate epiossification to the marginal crenulations of the parietal, which warrants testing within the phylogenetic analyses. One final character (109) was added to account for the degree of elongation of the nasal ornamentation ([0] as dorsoventrally tall as anteroposteriorly long at base; [1] nasal horncore typically 2.5 times greater or less in height than basal length; [2] nasal horncore typically greater than 2.5 times as long as basal length; [3] short nasal boss, not homologous to state 0); previous character matrices included such a character for supraorbital ornamentation but not for nasal ornamentation, which is not suitably characterized as simply 'non-pronounced or distinct' (character 20). Taxon ages were calibrated using the dates provided by Fowler [18].

In the Bayesian phylogenetic inference analysis, gamma rates parameter was used to allow for varying, non-uniform rates of evolution between taxa. The analysis was conducted for 20 000 000 generations and sampled every 1000 generations, with the first 5 000 000 generations (25%) discarded as burn-in (at which point the average maximum standard deviation of split frequencies was 0.01). All chains and runs converged upon a single consensus tree. The parsimony phylogenetic inference analysis was conducted in PAUP* 4.0b10 [21] following the parameters of Evans & Ryan [22], including a 1000 replicate random addition traditional search including tree bisection and reconnection (TBR) branch swapping holding 10 trees per replicate, followed subsequently with another round of TBR branch swapping.

A recent study [23] proposed that the type specimen of *Styracosaurus ovatus* (USNM 11869) falls within the range of variation of *Styracosaurus albertensis*, and is therefore a junior synonym of *S. albertensis*. However, the Holmes *et al.* [23] study conducts a cursory survey of individual character states which overlap between *S. ovatus* and *S. albertensis*, and infers a broad range of possible morphologies for the diagnostic parietal ornamentation of a single centrosaurine taxon. While our study agrees that the genus name '*Rubeosaurus*' is unnecessary, we retain *S. ovatus* for a number of reasons. USNM 11869 was collected from the uppermost Two Medicine Formation of Montana from the Landslide Butte area. While the precise stratigraphic location of USNM 11869 is unknown, this region of Two Medicine Formation exposure preserves radiometric dates about 800 000 years younger than the upper temporal range for *Styracosaurus albertensis* [18]. Holmes *et al.* [23] identify exceptions to the typical diagnosis of *Styracosarus albertensis* (e.g. lack of P1 is an exception to the typical condition of *S. albertensis*) which match the morphology of USNM 11869 for individual characters, but nonetheless remain outliers compared to specimens like the holotype of *Styracosaurus albertensis*. By contrast, features such as lack of P1 processes are the typical condition in the eucentrosaurans of the younger strata of the uppermost Two Medicine Formation, rather than the exception, and other character states such as degree of elongation of the P5 and P4 processes follow a strongly stratigraphic trend (discussed in our present study). This indicates that evolutionary trends may be responsible for these differences between specimens of *Styracosaurus albertensis* and the type specimen of *S. ovatus*, rather than variation within a single species. In order for *S. ovatus* and *S. albertensis* to be considered one taxon, this taxon must express both large P1 processes and completely absent P1 processes, large and hook-like P2 processes and diminutive P2 processes, and fully elongate P5 processes and minimally elongate P5 processes. Additionally, all factors such as ontogeny and stratigraphic placement are not accounted for in each specimen included in the discussion of *S. albertensis* variation, leaving ambiguity to the cause of the pattern of wide variation they describe. Variation within a single taxon would most meaningfully be invoked within a contemporaneous population, when temporal change is not a factor, and when controlling for other affecting factors such as ontogeny. Alternatively, our study demonstrates that through the stratigraphic record of the Two Medicine Formation, eucentrosaurans progressively reduce the degree of elongation of P5 and P4, and lack P1 processes; therefore, because *S. ovatus* comes from younger strata than *S. albertensis* and likewise appears to follow the trend of P1 elimination and P5 reduction, we entertain a

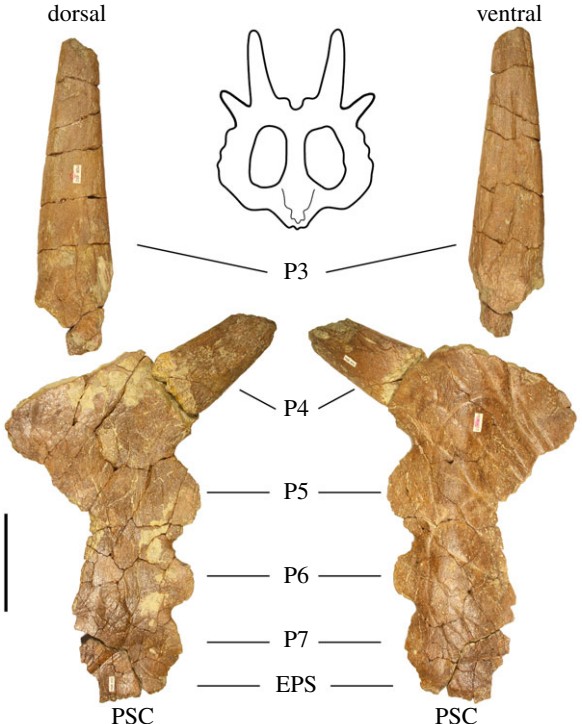

dorsal ventral

P3

P4

P5

P6

P7

EPS

PSC PSC

**Figure 4.** Left lateral parietal bar of *Stellasaurus ancellae* holotype MOR 492 in dorsal and ventral views. EPS, epiparietosquamosal; PSC, parietosquamosal contact. Scale bar 10 cm. Parietal line drawing modified from Evans & Ryan [22], Public Library of Science (PLoS), used under Creative Commons Attribution 4.0.

second working hypothesis and consider its morphology reflective of evolutionary change rather than variation within *S. albertensis*, and we retain *S. ovatus* as valid.

Regardless, the scope of this paper is reassessment of MOR 492 in relation to USNM 11869, regardless of what taxonomic designation is given to the latter specimen, and we demonstrate here that MOR 492 and USNM 11869 do not pertain to the same taxon (contra [13] and [14]), with MOR 492 representing a new taxon distinct from both *S. ovatus* and *S. albertensis*.

The electronic edition of this article and the nomenclatural acts contained in it have been registered in ZooBank, the online registration system for the International Code of Zoological Nomenclature and conforms to the requirements of the amended ICZN. The ZooBank LSID (Life Science Identifiers) for this publication is: urn:lsid:zoobank.org:pub:0F34F728-6FF2-4389-9F12-4EF961D1E4EA and can be viewed by appending the LSID to the prefix http://zoobank.org/. The electronic edition of this work was published in a journal with an ISSN, and has been archived and is available from the following digital repositories: PubMed Central, CLOCKSS and PORTICO. The new names published here are available under the ICZN from the electronic edition of this article.

# 4. Results

In order to reassess the previous referral of MOR 492 to *Styracosaurus ovatus*, we first describe its cranial anatomy while making as few inferences as to the homology of the critical parietal processes as possible. We compare the reassessed morphology of MOR 492 to previous interpretations of this specimen, and discuss the implications of our reassessment on the precise homology of the diagnostic parietal ornamentation, in particular, that MOR 492 is no longer referable to the taxon *Styracosaurus ovatus*.

## 4.1. Description

Parietal—MOR 492 preserves the left lateral bar of the parietal, contrary to McDonald & Horner [13] and McDonald [14] in which it was identified as the right lateral parietal bar, along with two broken parietal processes, one elongate and one partially elongate. The left lateral parietal bar (figure 4) preserves three marginal processes anteriorly, followed posteriorly first by the broken base of a partially elongate process

and finally by the broken base of an elongate process. The lateral bar is broken transversely immediately distal to the partially elongate process locus, with the base of the elongate process visible as a thickened cross section through the body of the parietal, and is otherwise complete to the parietosquamosal contact anteriorly. The anteriormost surface of the lateral margin, directly adjacent to the parietosquamosal contact, is punctuated by a small, rounded, bulbous mass of bone, identical to the epiparietal-squamosal (EPS) structures present in all other Two Medicine Formation centrosaurine specimens when preserved. McDonald & Horner [13] state that the squamosal contact of the left parietal bar of MOR 492 was not preserved, though it was later relocated by the first author as a separate fragment and reattached to the parietal. The dorsal and ventral surfaces of the parietal are identifiable by their respective typical ceratopsid surface textures, with the dorsal surface being semi-rugose with many small vessel indentations spanning the anteroposterior length of the lateral bar, while the ventral surface exhibits well developed but smoother bone than the dorsal surface, with many large, shallow vessel indentations, both spanning the length of the lateral bar and radiating posterolaterally from the left parietal fenestra. The dorsal surface is considerably flat with only two slight depressions in the transverse middle of the bar, while the ventral surface is shallowly convex laterally, giving way medially to a concave 'dish' approaching the fenestra, with a corresponding decrease in the dorsoventral thickness of the bar.

The anteriormost three processes adorning the lateral margin of the parietal are 'unmodified' (though see discussion for comments on the 'unmodified' condition of P5) tab-like crenulations, thinning to rounded apices with blade-like edges laterally. In dorsal and ventral views, these processes successively increase in size from locus to locus moving posteriorly along the lateral parietal bar, reminiscent of some *Styracosaurus* specimens and differing slightly from *Einiosaurus* and *Achelousaurus*, in which these processes are largely uniform in size; however, while slightly increasing in size, they are not elongate (see discussion for comments on the evolutionary changes affecting these processes). The anteriormost two processes are strongly imbricated, being inclined anteroventrally-posterodorsally in lateral view. This morphology is seen in all other non-fully mature Two Medicine Formation centrosaurine specimens (e.g. MOR 571; MOR 591; MOR 373-001; MOR 456 8-9-6-1; J. Wilson 2015, personal observation). Posterior to these three processes is a wide (8.3 cm) broken parietal process locus. MOR 492 was preserved with two broken, disassociated spikes, one small (13.9 cm in length) and one large (33 cm). The exact position of the smaller broken process has been debated, but can confidently be placed at this anteriormost broken locus based on a prominent, deep and narrow vessel indentation on the ventral surface which can be followed from the parietal bar across the break and onto the process, as noted by McDonald & Horner [13]. This spike is slightly curled such that it is concave dorsally and convex ventrally, unlike any of the elongate P4 spikes of *Einiosaurus* (e.g. MOR 373; locality TM-023), though here we do not consider this condition diagnostic. The longer spike is straight, with very slight dorsal curling matching that of the smaller spike, allowing for inference of dorsal and ventral surfaces. The dorsal surface is slightly mottled in texture and is largely devoid of any vessel impressions. This elongate process is inferred as corresponding to the posteriormost preserved process locus, based on the homology of this locus compared to other eucentrosaurans and its corresponding morphology (see below). The ventral surface, along with the medial edge, exhibits several shallow vessel impressions spanning the length of the spike.

McDonald & Horner [13] identify an abraded bone fragment as a frill epiossification and suggest that it may pertain to the EPS. Contrary to that assessment, we find that the bone fragment bears no features, such as the distinctive articular surface, which can be used to identify it as such. Rather, MOR 492 conspicuously preserves no frill epiossifications (epiparietals and episquamosals). None of the parietal processes bear morphological evidence of having been formed by fusion of epiparietals to the loci crenulations which they would cap. While this could be explained by complete fusion of the epiparietals to the parietal, there are likewise no frill epiossifications, either disarticulated or in the process of fusing to the frill, preserved in either of the two *Einiosaurus* bonebeds (MNI of 15 total), or with any of the four known individuals (including one juvenile, MOR 591 and one subadult, MOR 571) of *Achelousaurus* (J. Wilson 2015, personal observation). If each P2-P7 locus was capped with an epiparietal, each individual would possess 12 epiparietals, and therefore the minimum 15 *Einiosaurus* and four *Achelousaurus* individuals should account for a total of 228 epiparietals, yet zero are preserved. While this is not strictly evidence that the Two Medicine Formation centrosaurines lacked frill epiossifications, there is currently no fossil evidence that they did possess these structures. Furthermore, contrary to McDonald & Horner [13], there is no evidence that the EPS structures in the Two Medicine Formation centrosaurines were formed through fusion of an epiossification to the parietosquamosal contact, as in other ceratopsids like *Triceratops* [24] or *Centrosaurus apertus* (ROM 767) in which the EPS is a true epiossification which spans the parietosquamosal contact and bears articular surfaces for both the parietal and squamosal, but rather

dorsal          ventral

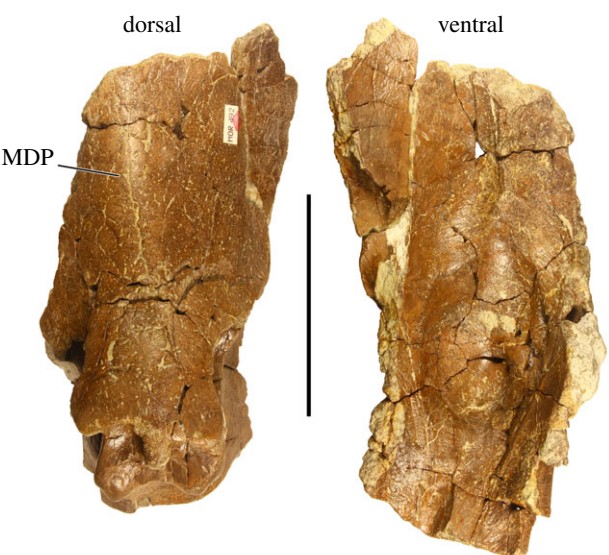

MDP

**Figure 5.** Anterior midline parietal bar of *Stellasaurus ancellae* holotype MOR 492 in dorsal and ventral views. MDP, midline dorsal prominence. Scale bar 10 cm.

appears to be outgrowths of bone relating to fusion of the parietal to the squamosals. The term 'EPS' is for now retained for clear reference to structures at analogous locations, though the term PSP, or 'parietosquamosal process', might be more appropriate for taxa in which the EPS-like structure has no evidence of being formed from a distinct epiossification. Given that the primary condition for Ceratopsidae is possession of epiparietals, we consider loss of frill epiossifications in the Two Medicine Formation centrosaurines to be the alternative hypothesis; however, this alternative working hypothesis must be currently favoured by the available specimens and evidence, and it is easily falsifiable by discovery of unambiguous epiparietals belonging to any of the Two Medicine Formation eucentrosauran taxa discussed here, in the form of disarticulated and unfused epiparietals, or parietals bearing epiparietals which are clearly in the process of fusing to the parietal.

MOR 492 additionally preserves the anterior end of the midline parietal bar (figure 5). The dorsal surface is rugose with numerous small vessel indentations and foramina punctuating the surface. The margins of the parietal fenestrae are broken, making it impossible to ascertain the width of the midline bar. The midline sagittal ridge is not strongly expressed, and remnants of one weakly defined midline dorsal prominence remain on the preserved portion of the bar, consistent with MOR 492 being relatively mature [25]. The ventral surface of the midline bar exhibits a deep depression opposite the articular frontal processes on the anterodorsal surface as in all centrosaurines.

Nasals—The nasals of MOR 492 are paired, fused and mostly complete, missing only the post-narial apron of the right nasal and a portion of the posterior margin of the nasal horncore, in addition to preservational damage to the apex of the horncore (figure 6). The anterior premaxillary processes are paired as well, though taphonomically broken from the nasals. Much of the bone surface is abraded and obscured on the right half of the nasal horncore, but the left half exhibits cortical bone which is rugose and punctuated by a number of vessel canals and associated foramina.

The nasal ornamentation is an erect, slightly recurved and highly elongate horncore, most closely resembling those of *Styracosaurus albertensis* (e.g. UALVP 52612; TMP 2005.12.58; TMP 1987.052.0001; CMN 344). The nasal horn is laterally compressed, with only slight lithostatic compression. The anterior margin of the base of the horn is slightly embayed, creating a slight overhang of bone, also seen in some specimens of *Styracosaurus* (e.g. UALVP 52612; TMP 1987.052.0001) and resembling the nasal overhang of *Coronosaurus brinkmani* [26] though with less pronounced osteological differentiation. As stated by Ryan & Russell [26], this feature likely corresponds to the keratinous sheath capping the horncore. While some specimens of *Centrosaurus apertus* express elongate nasal horns as well (e.g. AMNH FARB 5351), only *Styracosaurus albertensis* consistently expresses nasal ornamentation as hypertrophied as that of MOR 492. The nasal horncore of *Einiosaurus* holotype MOR 456 8-9-6-1 is likewise elongate, albeit highly procurved, but there are no immature specimens of *Einiosaurus* known with an elongate and erect nasal horncore (known juvenile specimens express erect horncores but none as elongate as specimens of *Centrosaurus*, *Styracosaurus* or MOR 492). Therefore,

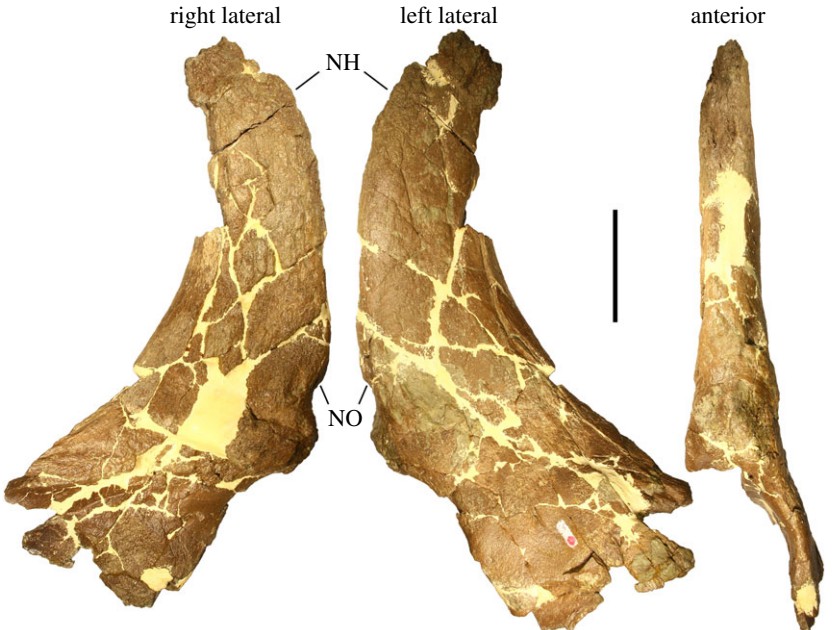

**Figure 6.** Nasal horncore of *Stellasaurus ancellae* holotype MOR 492 in right lateral, left lateral and anterior views. NH, nasal horncore; NO, nasal overgrowth. Scale bar 10 cm.

the exact developmental sequence of elongation versus procurvature in *Einiosaurus* is unknown, and it is unclear if *Einiosaurus* ever developed an elongate and erect nasal horncore similar to that of MOR 492 prior to procurvature.

Supraorbital ornamentation—MOR 492 preserves a partial left postorbital (figure 7). It is unclear if any of the palpebral is preserved; if so, it is fused imperceptibly, though if it was unfused the anterior extent of the preserved dorsal orbital rim is broken and abraded, obscuring the palpebral suture of the postorbital. Much of the lateral bone surface immediately dorsal to the orbit is damaged, though intact surfaces posterodorsal to the orbit and medial to the supraorbital ornamentation are rugose and mature in texture [25]. Although only the posterior half of the supraorbital ornamentation is preserved (including the posterior half on the apex), it can only plausibly be reconstructed as a short, diminutive horncore with a pointed apex, closely resembling the supraorbital ornamentation of all known *Styracosaurus albertensis* specimens and some *Centrosaurus apertus* [27,28]. The medial surface of the supraorbital horncore slopes gradually towards the skull midline. The medial surface of the postorbital is punctuated by three shallow corneal sinus depressions dorsal and posterior to the orbit. The near-posterior extent of the postorbital includes a partially preserved squamosal suture.

Premaxilla-MOR 492 preserves a partial left premaxilla, including the ventral approximately one-third of the rostral contact, the ventral angle, a portion of the premaxillary septum and the anterior approximately one-third of the posterior process of the premaxilla (figure 8), though centrosaurine premaxillae are generally not recognized as being diagnostic at the genus and species levels. The morphology of the preserved premaxilla does not differ significantly from closely related eucentrosaurans and therefore provides little additional taxonomic resolution.

## 4.2. Reassessment of 'Rubeosaurus' ovatus

As stated above, McDonald & Horner [13] inferred that the preserved left lateral parietal bar of MOR 492 would have been oriented in such a way that the P3 processes were medially inclined, allowing referral of MOR 492 to *Styracosaurus ovatus*. However, in their resulting phylogenetic assessment, *S. ovatus* did not form a sister taxon relationship with *Styracosaurus albertensis*, and thus the genus name *Styracosaurus* was replaced with *Rubeosaurus ovatus*, so as to avoid making the genus *Styracosaurus* paraphyletic [13]. McDonald & Horner [13] inferred the parietal homology of MOR 492 such that the elongate process was identified as P3, the partially elongate process as P4, and the three non-elongate processes were identified as P5, P6 and P7. It was noted that this homology produced a polymorphic condition for

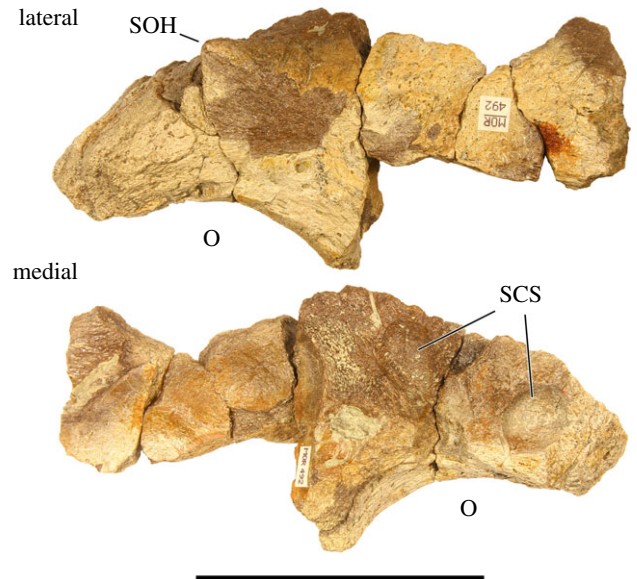

**Figure 7.** Left supraorbital ornamentation of *Stellasaurus ancellae* holotype MOR 492 in lateral and medial views. O, orbit; SCS, supracranial sinus; SOH, supraorbital horncore. Scale bar 10 cm.

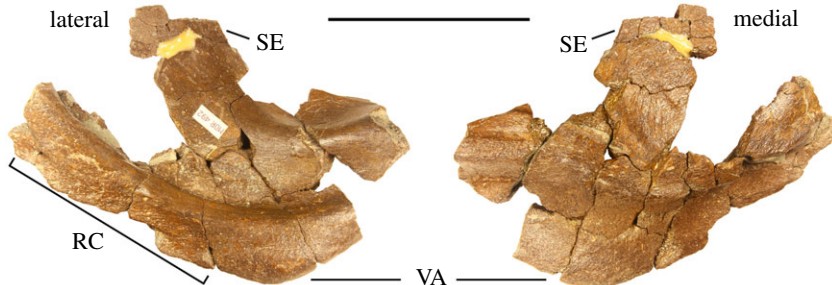

**Figure 8.** Partial left premaxilla of *Stellasaurus ancellae* holotype MOR 492 in lateral and medial views. RC, rostral contact; SE, premaxillary septum; VA, ventral angle. Scale bar 10 cm.

the P4 and P5 processes compared to the 'Rubeosaurus' ovatus holotype specimen USNM 11869, in which P4 is elongate like P3, and P5 is partially elongate.

McDonald [14] reassessed the parietal process homology of both MOR 492 and *Einiosaurus*, such that they expressed seven processes per side of the parietal rather than six, constituting P2–P8. In doing so, McDonald [14] noted that this gave MOR 492 a partially elongate P5 process, making it consistent with the morphology of the 'Rubeosaurus' ovatus holotype parietal, USNM 11869, but making the orientation of the P3 process ambiguous. In this revised interpretation by McDonald [14], the anteriormost parietal process of MOR 492 must be identified as P8 in order for the partially elongate process to be identified as P5, thus making it consistent with the 'Rubeosaurus' holotype despite the ambiguity of the P3 process orientation.

However, contrary to McDonald [14], no currently identified centrosaurines from the Two Medicine Formation express a P8 parietal process; eight parietal processes are only rarely encountered on any centrosaurine specimen, and when noted are inevitably discovered to be an improper coding for the presence of an epiparietosquamosal or its inflated locus, or atypical bilateral asymmetry. Instead, contrary to McDonald [14], the purported P8 process of *Einiosaurus* is an epiparietosquamosal-like structure which spans the parietosquamosal contact, rather than being a parietal marginal crenulation situated entirely on the parietal (figure 1). This EPS-like structure, which is also seen in MOR 492 and all other Two Medicine Formation centrosaurines which preserve this region of the parietal (e.g. MOR 456 8-9-6-1; MOR 456 8-29-87; figure 1), is usually dissimilar to the processes lining the margins of the parietal and squamosals, and is instead a bulbous mass of bone likely formed through initial fusion of the parietal to the squamosals. Because *Einiosaurus* is represented by a complete half parietal (holotype MOR 456 8-9-6-1), it unambiguously only expresses P2-P7 and an EPS. Furthermore, the processes identified as P8 in McDonald [14] represent entirely different structures in different specimens, with the

'P8' of *Einiosaurus* (MOR 456 8-9-6-1) being the EPS and the 'P8' of MOR 492 being the anteriormost parietal marginal crenulation.

Furthermore, the anteriormost two parietal processes of all Two Medicine Formation eucentrosaurans (with the exception of late-adult MOR 485 and an undescribed mature *Einiosaurus* skull (J. Wilson 2015, personal observation)) are imbricated, allowing for cross-identification of homologous parietal processes at the anterior end of the parietal of different specimens (figure 2). Like the *Einiosaurus* holotype MOR 456 8-9-6-1 and all other known Two Medicine Formation eucentrosauran specimens (except the two fully mature specimens noted above), MOR 492 expresses an EPS structure and imbrication of the anteriormost two parietal processes, making them identifiable as P7 and P6.

This makes referral of MOR 492 to *Styracosaurus ovatus* tenuous, as the anteriormost parietal process of MOR 492 must necessarily be identified as P8 for the partially elongate parietal process of MOR 492 to be identified as P5, which is necessary for MOR 492 to be congruent with USNM 11869. Most notably, because the orientation of P3 and the condition of P4 are ambiguous in that scenario, a partially elongate P5 would be the only character uniting MOR 492 to *Styracosaurus ovatus*. However, that scenario is only possible by applying two different homologies to identical structures on the anterior end of the lateral parietal bars (the two imbricated parietal crenulations; figure 3) of closely related eucentrosaurans from the Two Medicine Formation. Therefore, the simplest and most consistent parietal process homology for MOR 492 is an elongate P3, a partially elongate P4, and non-elongate P5, P6 and P7, plus an EPS structure. Therefore, the morphology of MOR 492 is not consistent with that of USNM 11869, and MOR 492 cannot be referred to *S. ovatus*.

Because MOR 492 is not referable to *Styracosaurus ovatus*, its additional cranial characters no longer pertain to the diagnosis of *Styracosaurus ovatus*. These additional cranial characters contributed to the revised phylogenetic placement of *Styracosaurus ovatus* which necessitated the erection of the new genus name '*Rubeosaurus*.' Additionally, McDonald [14] referred a juvenile centrosaurine specimen (USNM 14768) from the Two Medicine Formation to '*Rubeosaurus*,' but its morphology is likewise not consistent with the morphology of *Styracosaurus ovatus*. When viewed posteriorly, the proposed broken base of the right P3 of this specimen is a diminutive broken surface and is considerably narrower than the base of the corresponding left P3 process, making it unlikely to be the P3 locus. The left P3 process is additionally not medially inclined, as noted by McDonald [14], leaving no evidence for referral of this juvenile specimen to *Styracosaurus ovatus*. When the right lateral parietal bar of USNM 14768 is viewed laterally, the anteriormost two marginal crenulations are imbricated, identical to MOR 492 and MOR 456 8-9-6-1, thus allowing them to be identified as P7 and P6. In following, the more posteriorly positioned marginal processes can also be identified; the homology assigned by McDonald [14] is inaccurate, and the more consistent homology is that figured in fig. 10a of McDonald [14], matching the parietal homology of all other Two Medicine Formation eucentrosaurans. Therefore, with no specimens referable to *Styracosaurus ovatus* other than the holotype USNM 11869, the name '*Rubeosaurus*' *ovatus* becomes unnecessary, and this genus should be referred to by its original name, *Styracosaurus ovatus*. This is supported by the phylogenetic analysis reported here, in which *Styracosaurus ovatus* is once again recovered as the sister taxon to *Styracosaurus albertensis*.

## 4.3. Ontogenetic assessment of MOR 492

It is necessary to ensure that the ontogenetic status of MOR 492 does not bias interpretations of its morphology and thereby affect the outcome of phylogenetic analyses and other evolutionary assessments. The bone surface texture of all of the preserved cranial elements of MOR 492 is rugose and well developed, indicative of maturity [25,29]. Additionally, however, the degree of imbrication of the anteriormost two parietal processes is ontogenetically informative, with imbrication diminishing in the most mature adult individuals of *Einiosaurus* and *Achelousaurus* (J. Wilson 2015, personal observation). The combination of imbricated P7 and P6 processes and mature bone texture in MOR 492 is most closely comparable to *Einiosaurus* holotype MOR 456 8-9-6-1 and is thus likely indicative of a young adult individual. Unfortunately, MOR 492 does not preserve any elements appropriate for histological assessment, but the ontogenetic indicators previously established for centrosaurine ceratopsids [25,29] are consistent with the conclusion that MOR 492 was mature.

## 4.4. Phylogenetic analysis

In order to assess the phylogenetic placement of MOR 492 *Stellasaurus ancellae* gen. et sp. nov. within Centrosaurinae and in relation to the other centrosaurine taxa discussed here, we coded MOR 492 into

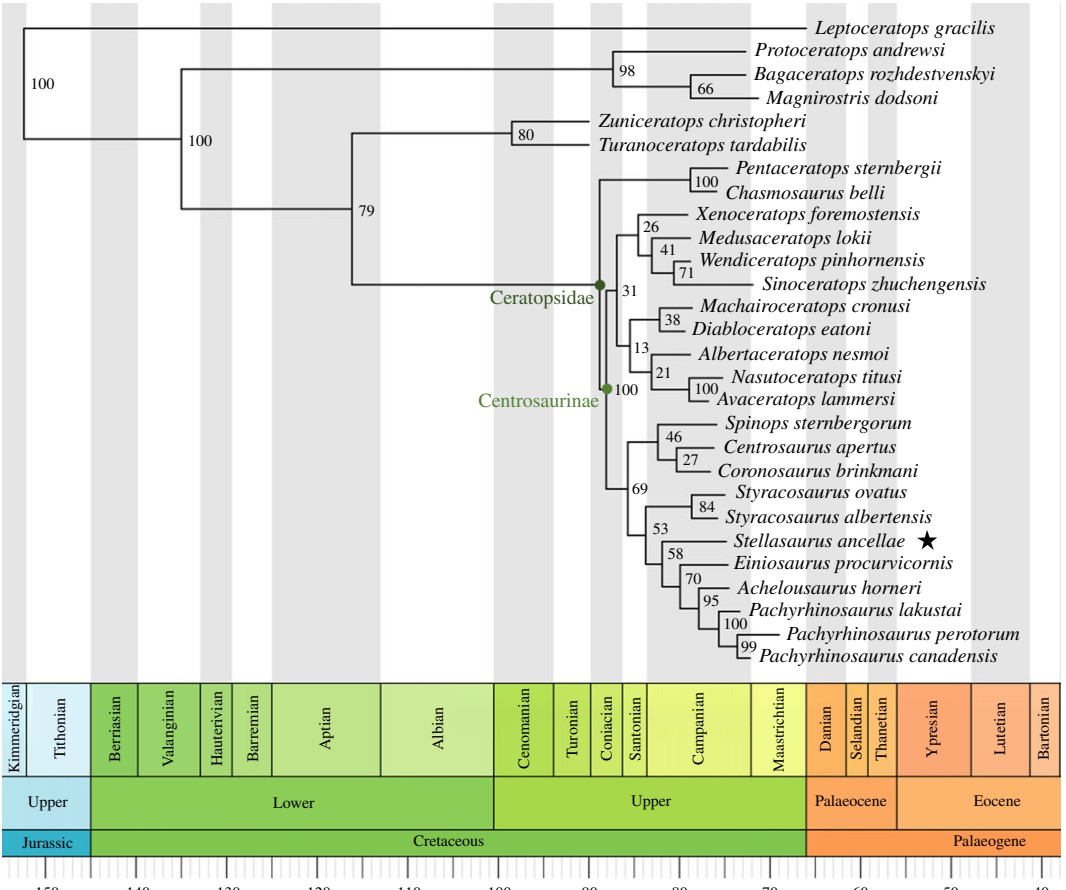

**Figure 9.** Consensus tree from Bayesian phylogenetic analysis, using a modified version of the matrix of [19]. The analysis was conducted in MrBayes 3.2 across four chains for 20 000 000 generations each, discarding the first 5 000 000 generations as burn-in. All chains converged upon the single consensus tree seen here. Posterior probabilities are noted at each node. Taxon ages occur at the end of the terminal branches and were calibrated from [18] as averages of any temporal range. *Stellasaurus ancellae* gen. et sp. nov. is indicated by a black star. Time scale produced using the strap package [30] for R [31].

the character matrix of Tykoski *et al*. [19] and conducted both a Bayesian (figure 9) inference analysis using the parameters of [6] and a parsimony inference analysis using the parameters of Evans & Ryan [22] (figure 10). A number of character states were modified for *Einiosaurus* to reflect its multi-state parietal P4 process expressions (see Discussion regarding stratigraphic trends within *Einiosaurus*). 'Rubeosaurus' was removed from the analysis, while *Styracosaurus ovatus* was coded solely based on the holotype specimen, USNM 11869. In addition, we added two characters to account for the degree of elongation of the P4 and P5 parietal processes (rather than just 'elongate' or 'non-elongate'); one character to test the hypothesized variable percentage of parietal processes represented by true epiossifications; and one character to account for variable elongation of the nasal horncore.

The Bayesian phylogenetic inference produced a resolved consensus tree, while the parsimony heuristic analysis produced a tree with large polytomies. In the parsimony tree, *Stellasaurus* is recovered in a large Eucentrosaura polytomy, while *Styracosaurus albertensis* and *Styracosaurus ovatus* are recovered as sister taxa. Pachyrhinosaurini is recovered similarly to the Bayesian analysis, though the three species of *Pachyrhinosaurus* are recovered as a polytomy. The parsimony analysis retained 129 trees, with the strict consensus tree having a length of 194 steps, CI = 0.5769 and RI = 0.7311. *Stellasaurus ancellae* is supported as a new taxon based on five unambiguous characters, including characters 23, 63, 102, 104 and 105. The resulting Bayesian phylogeny includes a number of notable aspects (figure 9). Reassessment of MOR 492 pulled *Styracosaurus albertensis* and *S. ovatus* from Centrosaurini [32] to the base of Pachyrhinosaurini, such that Centrosaurini is represented only by *Coronosaurus*, *Centrosaurus* and *Spinops*. The genus *Styracosaurus* is recovered as a sister taxon relationship between *S. albertensis* and *S. ovatus*, and is recovered intermediate to the Centrosaurini and Pachyrhinosaurini. *Stellasaurus* is recovered within Pachyrhinosaurini, as the sister taxon to the

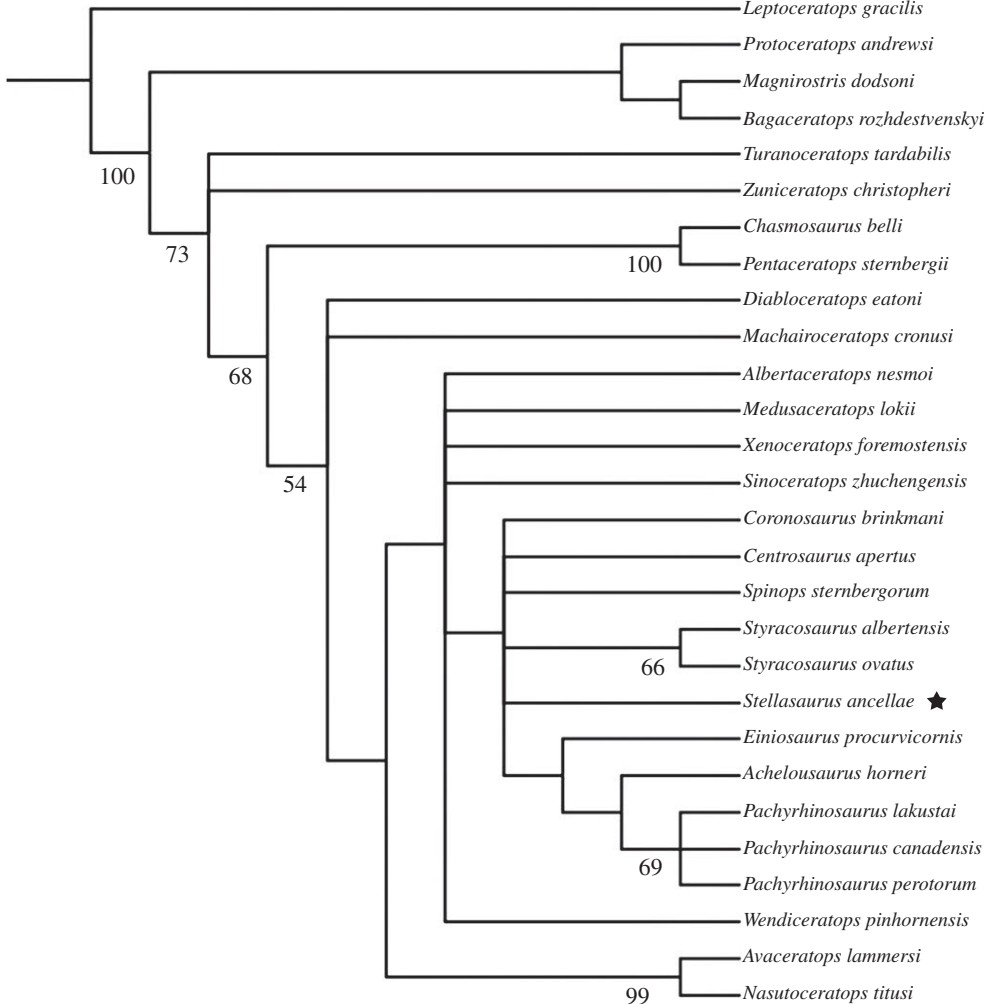

**Figure 10.** Strict consensus phylogeny of 129 most parsimonious trees (tree length = 194 steps, CI = 0.5769, RI = 0.7311) produced from parsimony heuristic search, with bootstrap values indicated. The analysis was conducted in PAUP* 4.0b10 using a modified version of the matrix of [19] and followed the parameters of Evans & Ryan [22], including a 1000 replicate random addition traditional search including TBR branch swapping holding 10 trees per replicate, followed subsequently with another round of TBR branch swapping.

least inclusive clade formed by *Einiosaurus*, *Achelousaurus* and *Pachyrhinosaurus*. Interestingly, the topology within Eucentrosaura is largely congruent with the stratigraphic occurrence of each taxon (although *Spinops* lacks precise locality and stratigraphic data), with the basal-to-derived tree construction following the stratigraphic occurrences of these taxa. The recovered relationships are, therefore, consistent with the hypothesis that a number of these taxa are related anagenetically, with the stepwise spine topology reflecting stratigraphic occurrence of these taxa. Posterior probabilities are lowest among the basalmost centrosaurines and higher in the derived taxa most prominently discussed here, though overall the posterior probabilities are consistent with previous Bayesian phylogenetic analyses of Centrosaurinae [6].

The sister taxon relationship between *S. albertensis* and *S. ovatus* may reflect a cladogenetic event, with *S. ovatus* branching off from the lineage which includes *S. albertensis* and the other Two Medicine Formation centrosaurines. Conversely, if it is a member of the lineage discussed here, it may simply indicate that the transformation of characters occurred in such a way that the character states of *S. ovatus* pull it most closely to *S. albertensis*. Unless *S. ovatus* is found to be contemporaneous with *S. albertensis*, which is unlikely given the overall younger age of the Landslide Butte area of the uppermost Two Medicine Formation compared to the upper Dinosaur Park Formation, the sister taxon relationship of *S. ovatus* and *S. albertensis* is not definitively reflective of a cladogenetic relationship between the two.

## 4.5. Systematic palaeontology

Dinosauria Owen, 1842
Ornithischia Seeley 1887
Ceratopsia Marsh, 1890
Neoceratopsia Sereno, 1986
Ceratopsidae Marsh, 1888
Centrosaurinae Lambe, 1915
*Stellasaurus* gen. nov.

### 4.5.1. Diagnosis

Monotypic, as for species
*Stellasaurus ancellae* gen et. sp. nov.

### 4.5.2. Etymology

The genus name *Stellasaurus*, 'star lizard', is derived from *Stella*, Latin for star, and –saurus, Greek for lizard, in reference to the overall star-like appearance of the cranial ornamentation, and in homage to the song 'Starman' by David Bowie. The species name *ancellae* honours Museum of the Rockies field palaeontologist and fossil preparator Carrie Ancell, who discovered and prepared MOR 492, the holotype specimen of *Stellasaurus ancellae*, as well as the holotype of *Achelousaurus horneri*, and co-discovered the holotype of *Einiosaurus procurvicornis*, and whose decades of extraordinary fossil preparation have furthered vertebrate palaeontology beyond measure.

### 4.5.3. Holotype

MOR 492, an isolated partial skull of a centrosaurine ceratopsid preserving the left lateral parietal bar, proximal portion of the midline parietal bar, near-complete paired and fused nasals, partial left premaxilla, and partial left postorbital and associated supraorbital ornamentation.

### 4.5.4. Locality, horizon and age

MOR 492 was recovered from Lithofacies 5 [17] of the uppermost Two Medicine Formation, 65 m below the upper contact between the Two Medicine Formation and the overlying Bearpaw Formation. The specimen was discovered in the Landslide Butte area, immediately adjacent to the US/Canadian border and 40 km northwest of the town of Cut Bank, MT ([15] fig. 2; [33] fig. 1). MOR 492 is bracketed stratigraphically by two radiometrically dated bentonites, one 55 m above (TM-6) and one 5 m below (TM-4) [18,34]. Fowler [18] recalibrates the sanidine radiometric date of TM-4 at about 75.03 Ma and the plagioclase date at 75.24 Ma, and recalibrates the plagioclase TM-6 date at 75.04 Ma. While this makes the stratigraphically higher TM-6 plagioclase date older than the stratigraphically lower TM-4 sanidine date, the plagioclase dates from both TM-4 and TM-6 are consistent stratigraphically. Fowler [18] also recalibrates a date from the Bearpaw Tuff, located 8 m above the contact between the Dinosaur Park Formation and the overlying Bearpaw Formation, at 75.46 Ma, making the strata from which MOR 492 was recovered contemporaneous with the final deposition of the Bearpaw Formation in Alberta and temporally younger than the Dinosaur Park Formation from which all known *Styracosaurus albertensis* specimens have been collected. This places the host stratum of MOR 492 immediately younger than 75.2 Ma.

### 4.5.5. Diagnosis

Centrosaurine ceratopsid exhibiting a unique combination of characters intermediate in distribution between the stratigraphically preceding *Styracosaurus albertensis* and stratigraphically successive *Einiosaurus procurvicornis*: elongate, erect and recurved nasal horncore and diminutive supraorbital ornamentation with pointed apex, as in *Styracosaurus albertensis*; parietal with elongate, straight P3 processes (spikes), partially elongate P4 processes (spikes) less than half as long as P3, and non-elongate P5, P6 and P7 processes, as in *Einiosaurus procurvicornis*; unique P4 elongation, intermediate between that of *Styracosaurus albertensis* and the stratigraphically successive lowest bonebed of *Einiosaurus*. As stated above, elongate, erect nasal horncores are variably present in *Centrosaurus apertus*, though rarely with the same recurvature and degree of hypertrophy as in both *Styracosaurus albertensis* and *Stellasaurus*. *Einiosaurus procurvicornis* specimens of

equivalent maturity (e.g. MOR 456 8-9-6-1; young adult, based on degree of P7 and P6 parietal process imbrication and bone surface texture) express strongly procurved nasal horns and rounded masses of bone as supraorbital ornamentation, differing significantly from the facial ornamentation seen in *Stellasaurus*.

## 4.6. *Styracosaurus ovatus* rediagnosis

### 4.6.1. Holotype

USNM 11869, an isolated partial parietal of a centrosaurine ceratopsid preserving the posterior parietal bar and partial midline and left lateral bars.

### 4.6.2. Locality, age and horizon

USNM 11869 was recovered from Lithofacies 5 [17] of the uppermost Two Medicine Formation [12]. The specimen was discovered in the Landslide Butte area, which is situated immediately adjacent to the US/Canadian border and 40 km northwest of the town of Cut Bank, MT ([15] fig. 2; [33] fig. 1). USNM 11869 lacks precise locality data [12,14] and is therefore of unknown precise stratigraphic placement within the Landslide Butte section. Though hypothetical, because USNM 11869 was recovered from the geographically constrained Landslide Butte area close to other sites with known stratigraphic data, its stratigraphic origin is likely not drastically different from the other centrosaurines from this area (*Achelousaurus*, *Einiosaurus* and *Stellasaurus*).

### 4.6.3. Diagnosis

Centrosaurine ceratopsid possessing one currently recognized autapomorphy [27]: medial inclination of the P3 parietal processes. Like *Styracosaurus albertensis*, *S. ovatus* expresses straight, elongate P3 and P4 processes, and partially elongate P5 processes. The preserved P5 process of the *S. ovatus* holotype specimen USNM 11869 is less elongate than the P5 process of *S. albertensis* holotype specimen CMN 344. *Styracosaurus ovatus* lacks P1 processes, unlike *S. albertensis* in which P1 possession is typical, but like the centrosaurines *Einiosaurus procurvicornis*, *Achelousaurus horneri*, *Pachyrhinosaurus lakustai*, *Pachyrhinosaurus canadensis* and *Pachyrhinosaurus perotorum*.

# 5. Discussion

Horner *et al*. [11] originally hypothesized that centrosaurine Transitional Taxon A (*Stellasaurus*; MOR 492) was a morphologically intermediate metaspecies that was descended from *Styracosaurus albertensis* and ancestral to *Einiosaurus* in an anagenetic lineage along with *Achelousaurus*. Conversely, Sampson [15] and Sampson *et al*. [25] argued for cladogenesis and faunal replacement to explain the sequence of taxa. To assess the significance of *Stellasaurus* within Centrosaurinae, including hypotheses regarding the evolutionary mode responsible for its origin and its relationship to other centrosaurine taxa, we evaluate it using Unified Frames of Reference [4]. Unified Frames of Reference attempts to holistically assess taxa (and specimens) within the context of stratigraphy, geography and phylogeny, while accounting for how the ontogenetic maturity and taphonomic preservation of specimens affect their interpretations. We consider what evidence is currently available for the discussed taxa, and ultimately with which hypotheses the evidence is most consistent. We address the vital issues of stratigraphy and ontogeny first, followed by discussion of the possible effects of geography on stimulating speciation, and finally phylogenetic implications.

## 5.1. Stratigraphy

With few possible exceptions (see below), there are a number of centrosaurine taxa (*Styracosaurus albertensis*, *Stellasaurus*, *Einiosaurus*, *Achelousaurus*, *Pachyrhinosaurus lakustai*, *Pachyrhinosaurus canadensis* and *Pachyrhinosaurus perotorum*) with clear stratigraphic separation and potentially no contemporaneous occurrences, and which critically exhibit morphologies which appear intermediate and may represent metaspecies defined only by unique combination of plesiomorphic and derived characters while lacking autapomorphies in relation to the taxa which bracket them stratigraphically (figure 11).

   *Styracosaurus albertensis* is known solely from the upper Dinosaur Park Formation of Alberta, with Ryan *et al*. [27] providing more detailed stratigraphic context for a number of specimens and bonebeds. Some specimens (e.g. CMN 344) of *Styracosaurus albertensis* exhibit the diagnostic parietal ornamental

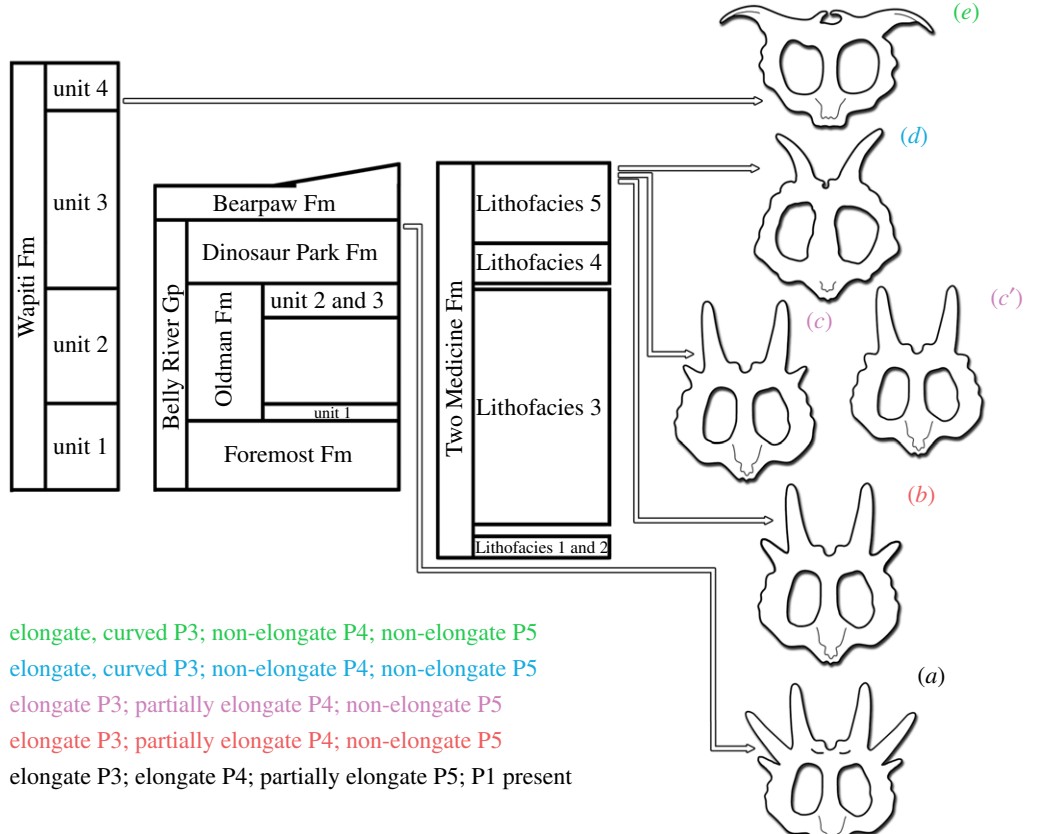

elongate, curved P3; non-elongate P4; non-elongate P5

elongate, curved P3; non-elongate P4; non-elongate P5

elongate P3; partially elongate P4; non-elongate P5

elongate P3; partially elongate P4; non-elongate P5

elongate P3; elongate P4; partially elongate P5; P1 present

**Figure 11.** Stratigraphic and temporal relationship of the centrosaurine taxa hypothesized as representing an anagenetic lineage, with the evolution of their parietal ornamentation, within their respective geological formations. Letters are colour coded by taxon to match transformation of parietal ornamentation listed. (*a*) *Styracosaurus albertensis*, (*b*) *Stellasaurus ancellae*, (*c*) and (*c′*) *Einiosaurus procurvicornis* MOR 373 and MOR 456, respectively, indicating slight stratigraphic separation with associated morphological change of the P4 process, (*d*) *Achelousaurus horneri*, (*e*) *Pachyrhinosaurus lakustai*. Generalized stratigraphic columns are based on [18]. Relative taxon placement is based on stratigraphic occurrence of specimens within their respective geological formations and the temporal relationships of host strata based on [18] (see Material and methods and Discussion). *Styracosaurus ovatus* is omitted due to its uncertain stratigraphic placement within the Two Medicine Formation. Parietal line drawings used and modified from Evans and Ryan ([22] fig. 15), Public Library of Science (PLoS), used under Creative Commons Attribution 4.0.

arrangement of elongate P3 and P4 processes and partially elongate P5 processes which are nearly as long as the P3 and P4 processes. These specimens constitute the majority of the known stratigraphic range of *S. albertensis* (Bonebed 42 to CMN 344). Other specimens variably express a short, tapered spike-like process or non-elongate crenulation at the P5 locus (e.g. ROM 1436). One of the stratigraphically highest occurrences of *S. albertensis*, large adult TMP 88.36.20 [27], expresses this combination of characteristically elongate P3 and P4 processes but a minimally elongate P5 process. There additionally appears to be a stratigraphic trend within *S. albertensis* of transverse narrowing of the posterior median embayment of the parietal, with this feature being prominently wide in stratigraphically low to median specimens (Bonebed 42 material; TMP 86.126.1) and narrowing in stratigraphically higher specimens (TMP 88.36.20; TMP 89.97.1), such that the highest specimens exhibit a posterior parietal midline embayment which is narrow and similar to that of *Einiosaurus* and *Achelousaurus*. Despite being only partially preserved, CMN 344, the holotype specimen of *S. albertensis*, appears somewhat intermediate among *S. albertensis* specimens in this regard, consistent with its median stratigraphic provenance within the upper Dinosaur Park Formation. With regards to the nasal and supraorbital ornamentation, there appears to be no stratigraphic change within *Styracosaurus albertensis*, with most specimens exhibiting the characteristic elongate, erect nasal horn and diminutive, pointed supraorbital horncores. Fowler [18] provides a radiometric date of 76.1 Ma from the Lethbridge Coal Zone, which constitutes the uppermost unit of the Dinosaur Park Formation. Therefore, the stratigraphically highest occurrences of *Styracosaurus albertensis* are slightly older than 76.1 Ma, predating the eucentrosauran centrosaurines from the Two Medicine Formation by about 1 Myr.

*Stellasaurus ancellae* is the stratigraphically lowest diagnostic (to the genus and species level) centrosaurine from the Two Medicine Formation with contextual stratigraphic data, deriving from about 65 m below the contact with the overlying Bearpaw Formation [11]. As established here, *Stellasaurus* maintains the elongate, erect nasal horn and diminutive supraorbital ornamentation of *Styracosaurus albertensis*, but exhibits further reduction in length of the P5 and P4 parietal processes. It expresses an elongate, straight P3, like *Styracosaurus albertensis*, but an only partially elongate P4 process and a non-elongate P5 process. It, therefore, exhibits shortening of the P4 and P5 processes relative to the temporally preceding *Styracosaurus albertensis*. As stated above, *Stellasaurus* was recovered from rocks which can be most closely dated to 75.2 Ma.

*Stellasaurus* is succeeded stratigraphically within the Two Medicine Formation by *Einiosaurus procurvicornis*, of which two bonebeds are known at 47 and 45 m below the overlying contact with the Bearpaw Formation [33]. Parietals from the stratigraphically lower bonebed (MOR TM-023) exhibit P4 processes which are intermediate in length between MOR 492 and the higher *Einiosaurus* bonebed (MOR TM-046). All parietals from TM-023 (MOR 373-001; MOR 373 6-28-6-4; MOR 373 7-9-87) bear P4 processes which are shorter than those of MOR 492 yet still partially elongate. Likewise, almost all parietals from TM-046 (e.g. MOR 456 8-29-87) express P4 processes which are slightly elongate, more so than the non-elongate P5, P6 and P7 processes, but less so than those of the TM-023 parietals. The only parietal from TM-046 with non-elongate P4 processes is the holotype of *Einiosaurus* (MOR 456 8-9-6-1), a young adult individual (J. Wilson 2015, personal observation). The trend of shortening parietal process P4 is therefore continued in *Einiosaurus*. Like *Stellasaurus*, *Einiosaurus* maintains straight, elongate P3 processes (until late in ontogeny, [35]). Juveniles of *Einiosaurus* possess short, erect nasal horncores (e.g. MOR 373 7-15-6-16; MOR 373 7-6-86-9; MOR 456 8-8-87-1), the typical juvenile centrosaurine condition, and short, pointed supraorbital horns like those of *Styracosaurus albertensis* and *Stellasaurus*, which develop through ontogeny into large, highly procurved nasal horns and rounded masses of bone dorsal to the orbits, respectively. It is currently unclear whether *Einiosaurus* first develops an elongate, erect nasal horncore like those of *Styracosaurus albertensis* and *Stellasaurus* which then procurves, or whether the short, erect nasal horn of juveniles develops directly into a large, procurved horn. Both bonebeds of *Einiosaurus* are bracketed stratigraphically by the same bentonites which bracket *Stellasaurus* MOR 492, but are 18 and 20 m higher in the formation than MOR 492, giving *Einiosaurus* a temporal range of about 75.2–75.1 Ma [18,34].

The centrosaurine with the highest stratigraphic occurrence within the Two Medicine Formation is *Achelousaurus horneri*, from 20 m below the contact with the overlying Bearpaw Formation [11]. *Achelousaurus* expresses fully reduced P4 parietal processes and elongate, laterally curved P3 processes [15]. The nasal and supraorbital ornamentation of *Achelousaurus* consist of high-ridged supraorbital bosses and a low nasal boss. *Achelousaurus* occurs 10 m below the highest sampled bentonite in the Two Medicine Formation [34], recalibrated by Fowler [18] to about 75.04 Ma, making it slightly older than this date.

*Pachyrhinosaurus* first occurs in the Wapiti Formation, represented by *Pachyrhinosaurus lakustai*, the type locality of which occurs 27 m below a dated volcanic ash with a recalibrated age of 73.7 Ma [18,36,37]. *Pachyrhinosaurus canadensis* occurs within the Drumheller and Horsethief Members of the Horseshoe Canyon Formation as well as the St Mary River Formation of Alberta [38,39]. *Pachyrhinosaurus perotorum* has a less well constrained stratigraphic occurrence within the Prince Creek Formation of Alaska [18,40]. All three species of *Pachyrhinosaurus* exhibit further modification of the cranial ornamentation, including more extensive lateral curvature of the P3 parietal processes compared to *Achelousaurus* and more robust nasal and supraorbital bosses.

The holotype of *Styracosaurus ovatus* is of unknown stratigraphic position within the Two Medicine Formation. However, because its parietal ornamentation is more similar to that of *S. albertensis* (elongate P3 and P4 processes, partially elongate P5 processes) than that of *Stellasaurus*, we hypothesize that it is from a stratigraphic position intermediate between *S. albertensis* and *Stellasaurus*, and below the latter in the local Landslide Butte section of the uppermost Two Medicine Formation.

Unfortunately, MOR 492 does not preserve either squamosal, and it is, therefore, unknown how many squamosal processes *Stellasaurus* would express. However, because it is stratigraphically intermediate between *Styracosaurus albertensis* and *Einiosaurus*, it is possible to hypothesize the number of crenulations each squamosal would bear. *Styracosaurus albertensis* is documented as possessing four to five processes per squamosal [27]. The stratigraphically lower of the two *Einiosaurus* bonebeds preserves five squamosals, four of which bear four processes plus EPS structures, while the stratigraphically higher *Einiosaurus* bonebed preserves six squamosals, five of which bear three marginal crenulations plus EPS structures. Because *Stellasaurus* occurs stratigraphically intermediately between *Styracosaurus albertensis* and the lower of the two *Einiosaurus* bonebeds (and only 18 m below the lower *Einiosaurus* bonebed) we predict that *Stellasaurus* would express four squamosal processes, plus EPS structures.

The stratigraphic separation and lack of documented contemporaneous occurrences among these taxa is critical when considering the evolutionary mode responsible for their origin. Anagenesis and cladogenesis, the two major evolutionary modes, are each inferable through different lines of evidence in the fossil record. Evidence may be consistent with hypotheses of anagenetic evolution (e.g. sequential, non-overlapping stratigraphic occurrences of closely related taxa which express intermediate morphologies), and anagenesis is falsifiable through discovery of overlapping stratigraphic occurrence of different taxa, whereas cladogenesis is demonstrable (through the same evidence which falsifies anagenesis) but is not strictly falsifiable (though individual scenarios may be inconsistent with cladogenesis) where no stratigraphic overlap between sister lineages occurs. Anagenesis is unfalsified and inferable in taxa that are stratigraphically sequential, demonstrate intermediate, transitional morphologies, and are of close geographical provenance. Taxa in the proposed phyletic series are metataxa and can be individually diagnosed by unique combinations of plesiomorphic and derived characters, and due to a lack of autapomorphies, do not represent different species from their ancestral internode [41]. Cladogenesis is the mode of evolution in taxa which are created by lineage splitting events, and in which taxonomic diversity is created, and is the sole possible evolutionary mode in fossil taxa which are contemporaneous and individually diagnosable. Geographical proximity of fossil occurrences represents a number of speciation scenarios, with likelihood of cladogenetic relationships becoming stronger over larger geographical ranges. There are numerous scenarios involving a cladogenetic relationship of taxa which may mimic the appearance of anagenetic evolution, but which involve more steps for which evidence must be provided. It should be remembered that anagenetic transformation of populations involves fewer steps than speciation of populations. Hypothesizing cladogenesis necessitates invoking speciation events (as speciation is defined as the 'origination or multiplication of species by subdivision' [42, p. 506], making it synonymous with cladogenesis), often facilitated by vicariance, in which an ancestral population is split, usually with some geographical barrier preventing gene flow and facilitating evolution in two separate daughter populations or a daughter population and an ancestral population. There is, therefore, more evidence which must be provided, and more steps to account for, to hypothesize that two taxa were produced through cladogenesis rather than anagenesis (figure 12). Gould [43] provided a standard for inferring cladogenetic punctuated speciation: 'We can distinguish the punctuations of rapid anagenesis from those of branching speciation by invoking the eminently testable criterion of ancestral survival following the origin of a descendant species. If the ancestor survives, then the new species has arisen by branching. If the ancestor does not survive, then we must count the case either as indecisive, or as good evidence for rapid anagenesis—but in any instance, certainly not as evidence for punctuated equilibrium.' This test relies upon the ancestral taxon persisting and remaining unchanged (thus allowing for it to be recognized as the ancestral taxon) and may be confounded if the daughter populations/taxa both experience evolutionary change. Regardless, the nature of cladogenetic speciation and the propositions made when inferring it necessitate the appropriate lines of evidence.

In considering the evidence available in the cases of the eucentrosauran centrosaurine taxa discussed here, there is no documented evidence of contemporaneous occurrences, making support of cladogenetic hypotheses currently tenuous (figure 12). There is likewise evidence of intermediate morphologies occurring in intermediate stratigraphic positions, which provides evidence for phyletic change through time. There is at least one specimen, an *Achelousaurus*-like pachyrhinosaur (TMP 2002.76.1) from the Lethbridge Coal Zone of Alberta [44], which may complicate a total-anagenesis scenario and affords a potential case of documenting cladogenesis through geographic partitioning. However, this specimen was collected from the bottom of a deep mud-filled valley that is incised into the Lethbridge Coal Zone, meaning it is likely much younger than its raw stratigraphic position suggests [44], and it may in fact be penecontemporaneous with *Achelousaurus* when new dates are generated. Within the Two Medicine Formation there are no eucentrosauran fossil occurrences which falsify the hypothesis that *Stellasaurus*, *Einiosaurus* and *Achelousaurus* represent intermediate members of an anagenetic lineage. Extending such hypotheses across geological formations affords more opportunity for documenting evidence of a falsifying nature, though the currently available evidence is consistent with the taxa discussed here being related through anagenesis.

## 5.2. Ontogeny

Because we infer that *Stellasaurus* holotype MOR 492 was approximately of equal maturity to the *Einiosaurus* holotype MOR 456 8-9-6-1, it, therefore, may be asked if the nasal horn and supraorbital ornamentation of MOR 492 may have developed *Einiosaurus*-like morphologies if the animal had lived longer. Late in ontogeny, *Einiosaurus* developed incipient *Achelousaurus*-like ornamental morphologies (e.g. MOR 456-1), and early in ontogeny, *Achelousaurus* (e.g. MOR 571) expresses morphologies seen in mature individuals

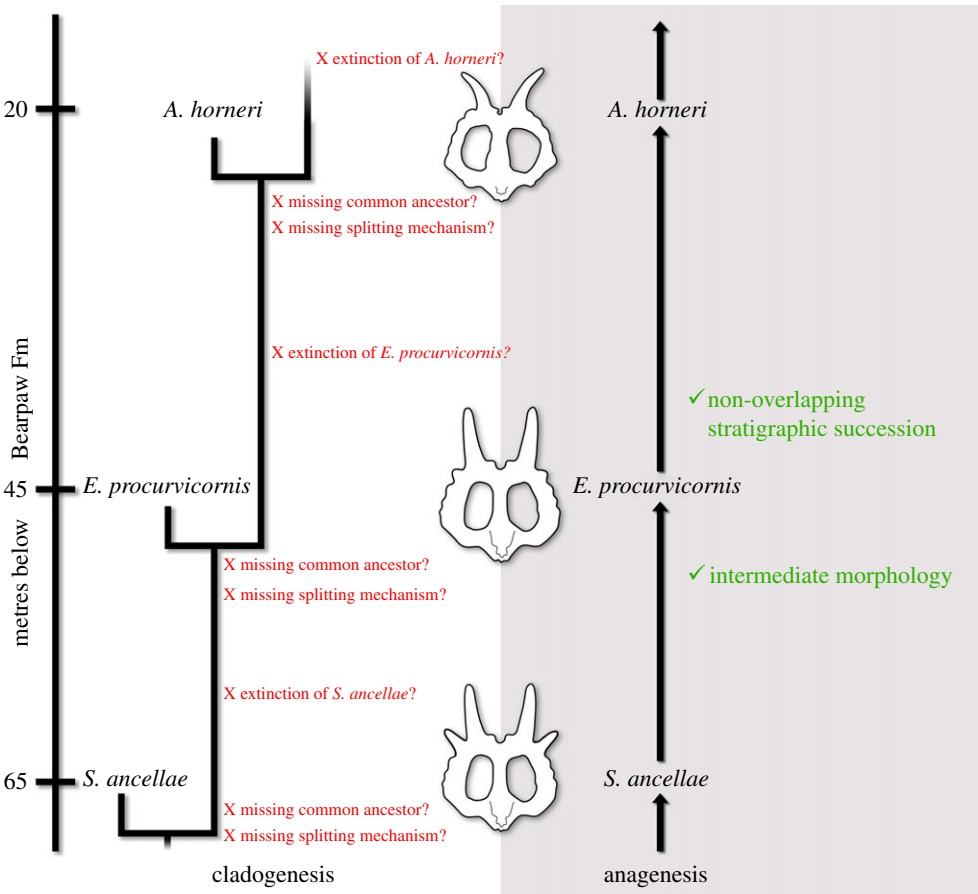

**Figure 12.** Comparison of the two major evolutionary mode hypotheses for the Two Medicine Formation centrosaurines and the associated lines of evidence supporting or absent. Parietal line drawings modified from Evans & Ryan ([22] fig. 15), Public Library of Science (PLoS), used under Creative Commons Attribution 4.0.

of *Einiosaurus* [35]. These trends tentatively indicate that development of the cranial ornamentation was driven by peramorphic heterochrony. Mature skulls of *Einiosaurus* bearing incipient *Achelousaurus*-like ornamentation indicate that these animals did not cease modification of the cranial ornamentation through life, and thus, in older individuals, *Stellasaurus* may have likewise expressed incipient *Einiosaurus*-like morphologies. However, based on the trends in *Einiosaurus* and *Achelousaurus*, these would have appeared considerably underdeveloped compared to the characteristic morphologies of *Einiosaurus*. A late-adult individual of *Stellasaurus* may, therefore, have begun to procurve its nasal horn and thicken its supraorbital ornamentation, whereas these features are already well-developed in subadult individuals of *Einiosaurus*. Recovery of more mature individuals of *Stellasaurus* than MOR 492 will test whether such trends occurred late in ontogeny in this taxon, though currently there is no evidence that *Stellasaurus* expressed morphologies similar to the diagnostic facial ornamentation of *Einiosaurus*.

There is currently no evidence that young adult individuals of *Einiosaurus* and *Achelousaurus* underwent further, extensive growth of the non-P3 parietal processes, instead only modifying the curvature of the P3 processes. It is, therefore, unlikely that the young adult MOR 492 would have undergone the extensive growth of the P4 and P5 processes necessary for it to resemble either *Styracosaurus ovatus* or *Styracosaurus albertensis*. Compared to these trends of ontogenetic development in *Einiosaurus* and *Achelousaurus*, it does not appear that the ornamental morphologies of MOR 492 and the interpretations of them were biased by the ontogenetic status of that individual.

## 5.3. Geography

All of the diagnostic eucentrosauran centrosaurine specimens from the Two Medicine Formation were recovered from a constrained geographical region in the Landslide Butte area of Montana (see [15] fig. 2; [33] fig. 1). This exposure of outcrop has maximum dimensions of about 6.5 by 9.61 km, though the actual area of the outcrop is about 22.5 km$^2$. While this does not imply that the total geographical range

of these taxa was only that large, the available evidence demonstrates that they inhabited the same geographical area through the time interval represented by their total stratigraphic range. This leaves few tenable hypotheses for the evolutionary mode responsible for these taxa. The stratigraphic separation and geographical ubiquity of these taxa fails to falsify, and is consistent with, the hypothesis that these animals represent members of an anagenetic lineage, with this scenario involving the least number of variables for which to account.

However, a number of alternative hypotheses involving cladogenetic relationships may be considered, though largely lack corresponding evidence currently (e.g. [25]). If these taxa had speciated cladogenetically, it would have likely necessitated extreme niche partitioning to support sympatric speciation, repeated speciation elsewhere and migration to the Landslide Butte area (perhaps involving extinction and replacement, as in the lineage turnover hypothesis [45]), or repeated sympatric speciation and extinction. The available evidence may be weighed against each of these hypotheses.

Extreme niche partitioning to facilitate sympatric speciation does not seem to be currently supported by morphological evidence. The available functional elements of *Einiosaurus* and *Achelousaurus* (numerous elements from MOR localities TM-023, TM-046, TM-060, TM-077 and TM-072), mainly the jaws and dentition, show little change and therefore no indication that these animals exploited different niches. Furthermore, the hypothesis of niche partitioning as a mechanism for sympatric speciation is only meaningful when applied to taxa which are contemporaneous; if, for example, the non-contemporaneous Two Medicine Formation eucentrosauran centrosaurines did exhibit different functional morphologies attributable to ecological adaptation, these differences would just as likely represent evolutionary responses of one lineage to a changing environment through time as they would represent niche partitioning within one unchanging ecosystem. Mallon & Anderson [46] found no morphometric evidence that individual taxa within the same subfamily (i.e. individual genera and species within Centrosaurinae) exhibited dietary niche partitioning (though as noted by Ryan *et al.* [32]) that study did not incorporate nasutoceratopsins, although that is inconsequential to the topic of niche partitioning within the Two Medicine Formation centrosaurines, which are not nasutoceratopsins).

There is currently no evidence for vicariance outside of and migration to the Landslide Butte area, or for repeated turnover events [45], which would take the form of both geographical barriers or habitat fragmentation stimulating allopatric speciation and contemporaneous occurrence of at least two of these taxa at any given time. Likewise, there are no known occurrences of *Stellasaurus*, *Einiosaurus* or *Achelousaurus* outside of the Landslide Butte area, and therefore no evidence of their presence elsewhere geographically, which would be necessary to support hypothetical vicariance and migration to the Landslide Butte area. Additionally, while the evidence for these scenarios may eventually be found, they involve numerous additional steps which produce more complicated conditions than are currently evident.

Repeated sympatric speciation (though noted for its complications above) and extinction of one daughter taxon could produce a stratigraphic and geographical pattern similar to the one produced by the current specimens and stratigraphic evidence, and would be demonstrable through repeated short intervals of contemporaneous occurrences of two individually diagnostic taxa (i.e. *Stellasaurus* and *Einiosaurus*, then *Einiosaurus* and *Achelousaurus*) followed by disappearance of one taxon, though this evidence does not currently exist, and further involves more steps than the simplest hypothesis. The morphology of each stratigraphically successive eucentrosauran taxon/site (e.g. both bonebeds of *Einiosaurus*) within the Two Medicine Formation is intermediate with regards to the taxa/sites which precede and succeed it, a pattern which would be possible cladogenetically, though requiring additional stimulus for speciation and extinction for which there is currently no evidence, but which is the pattern expected of anagenetic evolution.

The total inclusive geographical range of the Two Medicine Formation centrosaurines plus the geographical range of *Styracosaurus albertensis* and *Pachyrhinosaurus* is considerably greater, affording more opportunity for geographical segregation and speciation (especially, in the case of *Pachyrhinosaurus perotorum* from Alaska). In this regard, one major consideration is the westward extent of the Western Interior Seaway shoreline during the corresponding temporal interval, specifically whether or not the seaway abutted the Sevier thrust front, causing geographical isolation of inhabited land surfaces. Lillegraven & Ostresh [47] estimate palaeoshoreline for each ammonite biozone during the entire interval in which the centrosaurine taxa discussed here lived. Their results imply that no such westward seaway transgression abutting the Sevier thrust front occurred during this interval until the deposition of the Horseshoe Canyon Formation in Alberta, at which time little to no terrestrial space was left in Montana for the centrosaurines discussed here to inhabit. This is not to imply that localized incursions of the seaway on more brief timescales could not have occurred and stimulated speciation among these centrosaurines prior to this, yet the possible evidence for such scenarios is not yet known. Deposition of

the Dinosaur Park Formation and lower strata of the Two Medicine Formation than the Landslide Butte area occurred simultaneously, and therefore the inhabitable geographical range of *Styracosaurus albertensis* included both Montana and Alberta (although no evidence of *S. albertensis* occurring in the Two Medicine Formation is known). However, by the time the uppermost units of the Two Medicine Formation were being deposited, transgression of the interior seaway was depositing the Bearpaw Formation in south-central Alberta, constricting the inhabitable terrestrial geographical area of the Two Medicine Formation centrosaurines to Montana. Likewise, deposition of the Bearpaw Formation overlying the Two Medicine Formation in Montana occurred simultaneously with deposition of the Horseshoe Canyon Formation, constricting the geographical range of *Pachyrhinosaurus* to Alberta (*Pachyrhinosaurus* could potentially occur later in the St Mary River Formation of Montana as well, though there is not yet evidence of this, as the St Mary River Formation is not well studied). Therefore, transgression of the interior seaway shifted inhabitable terrestrial geographical space, but there is not currently evidence that it partitioned eucentrosauran populations. Such evidence would consist of geographically separated, contemporaneous, individually diagnostic taxa which arose through vicariance of a single population.

## 5.4. Taphonomy

Two taphonomic issues have complicated interpretations of MOR 492. Previous interpretations [13,14] regarding the orientation of the lateral parietal bar and the P3 process were influenced by the lack of a clear fit between the preserved, broken P3 process and its locus on the left lateral parietal bar, and the lack of the squamosal contact of the left lateral parietal bar. The fragment of the left lateral parietal bar bearing the squamosal contact was relocated by the first author and reattached to the lateral bar, clarifying the orientation of the parietal bar and likewise the orientation of the ornamentation it bears. As in parietals of *Einiosaurus* and *Achelousaurus*, the squamosal contact of MOR 492 is oriented transversely with little oblique tilt, unlike the parietal reconstruction of McDonald & Horner [13]. The tilted reconstruction of McDonald & Horner [13] in part contributed to the perception that the P3 processes of MOR 492 would have been medially inclined, thus making the specimen appear referable to *Styracosaurus ovatus*. The lateral parietal bars of MOR 492 and *Einiosaurus* are anteroposteriorly near-straight, which along with the transversely oriented parietosquamosal contact directs the P3 locus posteriorly rather than medially. In regards to the preserved, broken P3 process of MOR 492, it is most conservative to consider its orientation strictly ambiguous, though the parietal's overall near-identical morphology to that of *Einiosaurus* makes an *Einiosaurus*-like non-medially inclined orientation most conservative. MOR 492 is therefore further made non-referable to *Styracosaurus ovatus*.

## 5.5. Phylogeny

As would be expected by its intermediate stratigraphic position and unique combination of plesiomorphic and derived characters, *Stellasaurus* was recovered in an intermediate position between *Styracosaurus* and *Einiosaurus* in the Bayesian consensus time-calibrated phylogeny. The transformation of phylogenetic characters matches the stratigraphic succession of the eucentrosaurans from the Two Medicine Formation, and is consistent with their hypothesized recognition as metaspecies within a single, evolving lineage.

## 5.6. Evolution of socio-sexually selected cranial ornamentation

Among its closely related eucentrosaurans, *Stellasaurus ancellae* appears to mark an evolutionary shift in the specific cranial ornamental structures being acted upon most heavily by selection pressures. Centrosaurini is composed at its base of *Coronosaurus brinkmani* (Oldman Formation), *Centrosaurus apertus* (Oldman and Dinosaur Park Formations), *Spinops sternbergorum* (Oldman or Dinosaur Park Formation) and *Styracosaurus albertensis* (Dinosaur Park Formation) [22,32] succeeded temporally by *Stellasaurus ancellae* from the uppermost Two Medicine Formation. In these taxa, the facial ornamentation (nasal and supraorbital horncores) experiences little inter-taxon change with the exception of nasal horn elongation in *Styracosaurus* (which is maintained in *Stellasaurus*) and slight reduction in size and complexity of the supraorbital horncores between *C. brinkmani*, *C. apertus*, *Styracosaurus* and *Stellasaurus*. However, the parietal ornamentation of these taxa undergoes dramatic evolutionary change, ranging from the node clusters of *C. brinkmani*, to the curved, hook-like P1 and P2 processes of *C. apertus*, to the hypertrophied parietal processes of *Styracosaurus*. From this trend (aspects of which were noted by Ryan & Russell [26]), it can be inferred that selection pressures acted most heavily and intensely upon the parietal in these taxa (regardless of the specific evolutionary mode responsible) with less selection appearing to act on the

facial ornamentation. In this context, *Stellasaurus* represents the early stages of parietal simplification, seen in the shortening of the P4 and P5 processes and probable elimination of P1. *Stellasaurus* is also the last eucentrosauran to express the simple, erect (albeit hypertrophied) nasal horncore and diminutive supraorbital ornamentation. In the pachyrhinosaurins which succeed *Stellasaurus* temporally, the facial ornamentation undergoes dramatic evolutionary change, from the nasal horncore procurvature of *Einiosaurus* to the nasal and supraorbital boss development of *Achelousaurus*, which is modified further in *Pachyrhinosaurus*. In pachyrhinosaurins, the parietal ornamentation is markedly simpler than in the centrosaurins, with evolutionary change predominantly limited to further shortening and simplification of the parietal processes and curvature of the P3 processes (note the exception of parietal midline bar spike development in *Pachyrhinosaurus lakustai*) [48]. It can, therefore, be hypothesized that in the eucentrosauran taxa succeeding *Stellasaurus*, selection affected the facial ornamentation most heavily, with significantly less selection acting on the parietal. It is possible that the elimination of frill epiossifications in the Two Medicine Formation centrosaurines, noted above, occurred in conjunction with this overall trend of shortening and simplifying the parietal processes, though it must be noted that epiparietals occur in *P. lakustai* [48]. It is widely thought that the cranial ornamentation of ceratopsids functioned as socio-sexual signalling structures, with considerable debate surrounding specific hypotheses falling under sexual selection versus species recognition concepts (e.g. [49–54]). It appears that within eucentrosaurans, socio-sexual selective pressure switched from a parietal-dominated to a facial-dominated pattern, with *Stellasaurus* occurring at the earliest sequence of this evolutionary shift.

## 5.7. Occurrences of anagenetic evolution in dinosaurs

Evolutionary trends consistent with anagenesis have been hypothesized with increasing frequency in dinosaurs following Horner *et al*. [11], which explicitly inferred anagenesis as the evolutionary mode of four lineages of Cretaceous dinosaurs based on specimens from the Two Medicine Formation. The present study fails to falsify that hypothesis with the available evidence for the eucentrosauran centrosaurine ceratopsids originally documented by Horner *et al*. [11]. Evans [55] and Brink *et al*. [56] failed to reject anagenesis as the evolutionary mode relating *Hypacrosaurus stebingeri*, the hadrosaur metaspecies documented in Horner *et al*. [11], to *H. altispinus*. Brink *et al*. [57] later identified an autapomorphy of *H. stebingeri*, which would possibly falsify the hypothesis that *H. stebingeri* represents a transitional taxon; however, they shortly after state that this autapomorphic character could itself be 'evolutionarily labile and change from a *Corythosaurus*- or *Lambeosaurus*-like ancestor through *H. stebingeri* to *H. altispinus* in an anagenetic lineage through time' [57, p. 259]. Carr *et al*. [58] also supported the hypothesis of Horner *et al*. [11] that the tyrannosaurine taxon from the Two Medicine Formation, *Daspletosaurus horneri*, is consistent with being anagenetically related to *Daspletosaurus torosus*, with *D. horneri* from the uppermost Two Medicine Formation succeeding *D. torosus* from the Dinosaur Park Formation (following a similar pattern of stratigraphic occurrence to the centrosaurines discussed here). Scannella *et al*. [59] placed a large number of specimens of *Triceratops*, the most commonly recovered chasmosaurine ceratopsid, in stratigraphic context, which revealed that *Triceratops horridus* and *Triceratops prorsus* are stratigraphically sequential, with a transitional form occurring temporally intermediately. Freedman Fowler & Horner [60] describe a brachylophosaurin, *Probrachylophosaurus bergei*, with intermediate development of the nasal crest from an intermediate stratigraphic position between *Acristavus* and *Brachylophosaurus*, the basis for the inference that these taxa constitute an anagenetic lineage. Arbour & Evans [61] discuss the possibility that *Zuul crurivastator*, an ankylosaurine from the upper Judith River Formation of Montana (upper Dinosaur Park Formation equivalent), represents an intermediate taxon within an anagenetic lineage, succeeding *Dyoplosaurus* from the lower Dinosaur Park Formation and being directly ancestral to *Scolosaurus* from the upper Two Medicine Formation. The hypothesis of Arbour & Evans [61] is therefore similar to that of the centrosaurines discussed here and with that of Carr *et al*. [58], involving anagenetic lineages formed by taxa from the Dinosaur Park Formation (or its equivalents in the Judith River Formation) and the uppermost Two Medicine Formation. Recognition of multiple possible phyletic lineages of ornithischian dinosaurs spanning the Dinosaur Park and Two Medicine formations bolsters the hypotheses of anagenetic evolutionary change in each of these cases. Fowler [18] presents a holistic correlation of the terrestrial formations of the Cretaceous Western Interior of North America along with the stratigraphic occurrences of the ceratopsian, hadrosaurid and pachycephalosaurid taxa they produce, hypothesizing that the numerous resulting stratigraphic stacks of taxa may represent anagenetic lineages within these groups. In all of these cases, it is likely that continuous sampling and high-resolution stratigraphic data have contributed to these inferred trends, and, as recognized by Carr *et al*. [58], that the appearance of cladogenesis may in some instances be the result of insufficient sampling of the fossil record.

Because *Stellasaurus* is consistent with being a metataxon uniting *Styracosaurus* and *Einiosaurus*, it is in turn, consistent with being a transitional taxon within an evolving phyletic lineage with *Achelousaurus* and possibly *Pachyrhinosaurus*. While these taxa comprise the Pachyrhinosaurini, their hypothesized direct descent from *Styracosaurus* within a phyletic lineage appropriately makes them notable as the 'Styracosaurus-line' centrosaurines, denoting the earliest-occurring recognizable taxon within the hypothesized phyletic lineage.

## 5.8. Diversity of centrosaurine ceratopsids

The description of *Stellasaurus* increases the number of centrosaurine taxa from the Two Medicine Formation from three to four. However, there is currently no evidence that any of these taxa were contemporaneous; therefore, the diversity of Two Medicine Formation centrosaurines may be as few as one. *Styracosaurus ovatus* is of ambiguous stratigraphic position, and therefore ambiguous evolutionary origin, thus allowing the current potential diversity of Two Medicine Formation centrosaurines to be two. Recovery of additional specimens of *S. ovatus* with stratigraphic data will further resolve this issue. Following the assessments made here, if *Styracosaurus* and *Pachyrhinosaurus* form a single phyletically evolving lineage with *Stellasaurus*, *Einiosaurus* and *Achelousaurus*, they likewise contribute no additional taxonomic diversity, and suggest a much lower standing diversity of Centrosaurinae in the upper Campanian (and possibly into the lower Maastrichtian) of North America.

There are no known members of the Nasutoceratopsini clade which are contemporaneous with *Stellasaurus* or the other eucentrosaurans from the uppermost Two Medicine Formation. *Nasutoceratops titusi*, from the Kaiparowits Formation of Utah, is the youngest nasutoceratopsin and predates the Two Medicine Formation centrosaurine taxa by about 400 000 years [18]. If a member (or members) of this clade survived further in time so as to coexist with the uppermost Two Medicine Formation centrosaurines, the standing centrosaurine diversity would be a minimum of two lineages in the Upper Campanian of North America.

The highest diversity of centrosaurines appears to have occurred in the Middle Campanian, with nine taxa either coexisting or which occur in close succession but have not yet been assessed within direct ancestor–descendant relationships. The lower Oldman Formation of Alberta and equivalent units of the Judith River Formation of Montana have produced material currently referred to *Wendiceratops pinhornensis*, *Albertaceratops nesmoi* and *Medusaceratops lokii* [9,22,62], while the middle Oldman Formation of Alberta has produced *Coronosaurus brinkmani* [26] and the upper Oldman Formation has produced *Centrosaurus apertus* as well as material of a nasutoceratopsin [32,63]. Immediately below the Oldman Formation, the Foremost Formation of Alberta has produced the remains of *Xenoceratops foremostensis* [64], the oldest centrosaurine from the northern portion of the Western Interior of North America. Approximately coeval with these taxa from Alberta and Montana are *Diabloceratops eatoni* and *Machairoceratops cronusi* from the Wahweap Formation of Utah [5,8]. *Centrosaurus apertus* is predominantly known from the lower Dinosaur Park Formation and the time equivalent part of the Oldman Formation [63], immediately preceding and not overlapping stratigraphically with *Styracosaurus albertensis*. *Spinops* is of unknown precise stratigraphic placement but is thought to derive from the upper Oldman Formation or lower Dinosaur Park Formation [65].

Overall, centrosaurines, alongside their chasmosaurine relatives, exhibit an extreme diversity of ornamental cranial morphologies. *Stellasaurus ancellae* represents yet another member of this clade, which is ever-growing thanks to continued sampling of the terrestrial Cretaceous Laramidian formations. This wealth of disparate morphologies is evidence of strong, persisting socio-sexual selection pressures, often facilitated through dramatic ontogenetic development [48,66]. For the roughly 15 Myr for which a ceratopsid fossil record exists, there is almost no evidence of stasis within a taxon [32], meaning that unsampled or undersampled stratigraphic intervals are likely to produce new ceratopsid taxa, with some morphologies potentially being predictable and others surprising.

Ethics. MOR 492 was collected by the Museum of the Rockies from privately owned land with landowner permission, and is permanently accessioned into the collections of the Museum of the Rockies.
Data accessibility. The character list and character matrix used for the phylogenetic analyses performed in this study are provided here as electronic supplementary information.
Authors' contributions. J.P.W., M.J.R. and D.C.E. conceived of the research, collected data, designed and performed the experiments and wrote and edited the manuscript. All authors gave final approval for publication.
Competing interests. The authors declare no competing interests.
Funding. No funding was provided for this study.
Acknowledgements. Palaeontology is not possible without fossils, and therefore the people who find, excavate and prepare the fossils we study deserve the utmost respect and recognition. C. Ancell discovered MOR 492, as well as the holotype specimen of *Achelousaurus* and co-discovered the type locality of *Einiosaurus*, and her skilled preparation of these

specimens and countless others has facilitated decades of palaeontological research. J. Horner discovered both bonebeds of *Einiosaurus*, including co-discovery of the site which produced the type specimen. B. Harmon discovered subadult specimens of *Achelousaurus* which give critical insight into its ontogeny and ancestry. J. Horner and B. Makela put in motion and carried out one of the most productive field programmes in the history of palaeontology. We thank these people, along with everyone who discovered, excavated and prepared the other specimens discussed here, along with M. Weatherwax and the mosasaur specimen which first gained J. Horner access to the Blackfeet Reservation. We are grateful to the late G. Sundquist who kindly allowed the Museum of the Rockies to collect the holotypes of *Stellasaurus* and *Einiosaurus* from her land, along with the Blackfeet Nation for access to their lands. D. Dufault previously produced the parietal line illustrations which were used and modified here. K. Surya formatted figure 9. We thank J. Horner and J. Scannella for access to specimens in the MOR collection. J.P.W. thanks D. Tanke for photos and discussion of specimens in the collection of the RTMP. J.P.W. especially thanks K. Fowler for computer repair. Thanks to J. Gardner, H. Flora, K. Surya, J. Horner and especially D. Fowler for helpful discussion. J.P.W. thanks J. Scannella for mentorship in searching for J. Horner's metaphorical 'loose ends' in the scientific 'ball of twine.'

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
