## [Reviewer comments · Royal Society Open Science]

Review History

RSOS-200284.R0 (Original submission)

Review form: Reviewer 1 (Elizabeth Freedman-Fowler)

Is the manuscript scientifically sound in its present form?

Yes

Are the interpretations and conclusions justified by the results?

Yes

Is the language acceptable?

Yes

Do you have any ethical concerns with this paper?

No

Have you any concerns about statistical analyses in this paper?

No

Recommendation?

Accept with minor revision (please list in comments)

Comments to the Author(s)

I have a series of very minor, mainly technical edits that should be relatively quick to complete:

Ethics statement: These fossils came from someone's land, and either needed permits or permissions that should be mentioned here.

Abstract, lines 14-16: "currently considered valid" - Be careful with the phrasing here. Holmes et al 2020 refer *R. ovatus* to *Styracosaurus*. Of course you discuss this later in the paper, but in the abstract please use wording that everyone can agree on. Perhaps "herein considered valid", or "previously considered valid"?

Introduction, paragraph 1, line 22: add "larger nasal horns" to the centrosaur list to balance the chasmosaur list.

Introduction, paragraph 2: "currently recognized" - Same phrasing issue with *Rubeosaurus* as in the abstract.

Introduction: Excellent explanation of Unified Frames of Reference.

Institutional abbreviations: D.C. is missing a period.

Materials and Methods: Phylogeny paragraph is too long and should be separated into smaller paragraphs. Also, it includes some results (posterior probabilities) that should be moved out of here. PAUP should have a citation: Swofford, D. L. 2003. PAUP*. Phylogenetic Analysis Using Parsimony (*and Other Methods). Version 4.0b10. Sinauer Associates, Sunderland, Massachusetts. MrBayes should have a source citation as well.

Description, Parietal: I agree that this is a left lateral bar, not a right lateral bar.

Description, Parietal, second paragraph: "This morphology is seen in all other non-fully mature..." - add some specimen numbers to the parentheses (e.g. MOR xxx, MOR xxx; J. Wilson pers. obs.).

Description, Parietal, fourth page: "The term "EPS" is for now retained for clear reference to structures at homologous locations" - should this be analogous instead of homologous?

Phylogenetic analysis, first paragraph: "we added two characters to account for the degree of elongate" - elongate should be elongation, followed by a colon instead of a comma.

Taxon name: Beautiful.

Styracosaurus ovatus rediagnosis: "Monotypic, as for species" - does not apply here; you are trying to say that there are two species of *Styracosaurus*.

Discussion, Stratigraphy: Gould (2002) quote - include page number of quote.

Discussion, Stratigraphy: "an *Achelousaurus*-like pachyrhinosaur from the Lethbridge Coal Zone of Alberta (Ryan et al., 2010)" - Does this have a specimen number? "This specimen sits in the bottom" - Was it collected? Shouldn't use present tense.

Discussion, Ontogeny, second sentence: Because Wilson et al 2015 is just an abstract, include the specimen numbers that provide this ontogenetic evidence.

Discussion, Geography, second page: “localities TM-023; TM-046...” – add MOR before “localities”.

Discussion, Taphonomy: Well described.

Discussion, Occurrences of anagenetic, second page: “Freedman Fowler” preferably has a space, not a hyphen.

Discussion, Diversity of Centrosaurine Ceratopsids, second sentence: “any given point in time maybe be” – change to “may be”.

Figure 1 caption: If the structures labeled EPS are not thought to be distinct epiossifications, then shouldn't they be analogous rather than homologous?

Figures 1 & 2: Because two of the three bones in Figure 2 are the exact same photos as Figure 1, I suggest combining figures 1 and 2 to avoid duplication. Possibly Figure 3 could be combined into these as well, perhaps by using color or italics to differentiate the McDonald vs New numbers.

Figure 4: I understand the reasoning for having dorsal on the left and ventral on the right, so that the P# labels can be shared, but visually the figure would be much more intuitive if these were swapped left and right, because then the whole frill outline would be clear. The P# labels can be duplicated on each side. Also, the current font size of the P# labels seem huge. Looks like this is currently planned for full page width? One-column width might be enough, depending on whether just the shape outline matters, or whether you really need to see details of the bone texture.

Figure 6: I'd be interested in seeing an anterior view as well, to note the thickness (width) of the horncore.

Figures 9 & 10: Need a lot more information in the caption. Which matrix was used? Details of the methods? RI, CI? Which consensus tree is it, out of how many? I assume those are bootstrap numbers, but it needs explanation in caption. Also need to credit source of colored timeline, unless you made it completely from scratch.

Figure 9: Which point is the age of each taxon? Where the line ends? Where the name starts? What about uncertainties or ranges? You really don't need the late Paleogene taking up space. Trim it, so you have room to stretch out the time periods that matter. The time scale doesn't have to be perfectly linear the whole way – certain areas can be compressed or stretched, as long as it is clearly marked.

Figure 11: What is on the y-axis? Time? Are the stratigraphic placements of taxa relative or absolute? Are the specimen placements based on Fowler 2017 or other data? Where does *ovatus* fit in? You mention in the methods that the exact strat is unknown, but you do know that it's from the Landslide Butte area, which narrows the range- is it enough to plot on this figure? If *ovatus* can't be plotted, say so in the caption.

Figure 12: Excellent. I really like the clear way this shows all of the missing evidence for cladogenesis, and the simple, parsimonious anagenesis hypothesis. This figure is a great “visual abstract”.

Review form: Reviewer 2 (Julie Reizner)

Is the manuscript scientifically sound in its present form?

Yes

Are the interpretations and conclusions justified by the results?

Yes

Is the language acceptable?

Yes

Do you have any ethical concerns with this paper?

No

Have you any concerns about statistical analyses in this paper?

No

Recommendation?

Accept as is

Comments to the Author(s)

This more comprehensive approach to a species description is necessary in light of the complex (and largely unknown) challenges that ontogenetic changes in dinosaurs, particularly the ceratopsians, present to naming new species. The in-depth discussions on ontogeny, evolutionary trends, and stratigraphy are much appreciated in this case to supporting a new species. I wonder if, when considering the small sample sizes we see with many of these ceratopsians, anatomical variation among individuals can be significant enough to appear to represent more species than are actually present? Maybe someday we'll find out!

The species epithet is a very nice gesture and Carrie certainly deserves it - good thinking there.

Decision letter (RSOS-200284.R0)

27-Mar-2020

Dear Mr Wilson,

On behalf of the Editors, I am pleased to inform you that your Manuscript RSOS-200284 entitled "A new transitional centrosaurine ceratopsid from the Upper Cretaceous Two Medicine Formation, Montana and the evolution of the 'styracosaur' dinosaurs" has been accepted for publication in Royal Society Open Science subject to minor revision in accordance with the referee suggestions. Please find the referees' comments at the end of this email.

The reviewers and handling editors have recommended publication, but also suggest some minor revisions to your manuscript. Therefore, I invite you to respond to the comments and revise your manuscript.

- Ethics statement

- Data accessibility

If you wish to submit your supporting data or code to Dryad (<http://datadryad.org/>), or modify your current submission to dryad, please use the following link:
<http://datadryad.org/submit?journalID=RSOS&manu=RSOS-200284>

- Competing interests

- Authors' contributions

- Acknowledgements

- Funding statement

Because the schedule for publication is very tight, it is a condition of publication that you submit the revised version of your manuscript before 05-Apr-2020. Please note that the revision deadline will expire at 00.00am on this date. If you do not think you will be able to meet this date please let me know immediately.

To revise your manuscript, log into <https://mc.manuscriptcentral.com/rsos> and enter your

Author Centre, where you will find your manuscript title listed under "Manuscripts with Decisions". Under "Actions," click on "Create a Revision." You will be unable to make your revisions on the originally submitted version of the manuscript. Instead, revise your manuscript and upload a new version through your Author Centre.

If your manuscript is newly submitted and subsequently accepted for publication, you will be asked to pay the article processing charge, unless you request a waiver and this is approved by Royal Society Publishing. You can find out more about the charges at <https://royalsocietypublishing.org/rsos/charges>. Should you have any queries, please contact openscience@royalsociety.org.

Best regards,

on behalf of Professor Marcelo Sanchez (Associate Editor) and Jon Blundy (Subject Editor)
 openscience@royalsociety.org

Reviewer comments to Author:

Reviewer: 1
 Comments to the Author(s)

I have a series of very minor, mainly technical edits that should be relatively quick to complete:

Ethics statement: These fossils came from someone's land, and either needed permits or permissions that should be mentioned here.

Abstract, lines 14-16: "currently considered valid" – Be careful with the phrasing here. Holmes et al 2020 refer *R. ovatus* to *Styracosaurus*. Of course you discuss this later in the paper, but in the abstract please use wording that everyone can agree on. Perhaps "herein considered valid", or "previously considered valid"?

Introduction, paragraph 1, line 22: add "larger nasal horns" to the centrosaur list to balance the chasmosaur list.

Introduction, paragraph 2: "currently recognized" – Same phrasing issue with *Rubeosaurus* as in the abstract.

Introduction: Excellent explanation of Unified Frames of Reference.

Institutional abbreviations: D.C. is missing a period.

Materials and Methods: Phylogeny paragraph is too long and should be separated into smaller paragraphs. Also, it includes some results (posterior probabilities) that should be moved out of here. PAUP should have a citation: Swofford, D. L. 2003. PAUP*. Phylogenetic Analysis Using Parsimony (*and Other Methods). Version 4.0b10. Sinauer Associates, Sunderland, Massachusetts. MrBayes should have a source citation as well.

Description, Parietal: I agree that this is a left lateral bar, not a right lateral bar.

Description, Parietal, second paragraph: "This morphology is seen in all other non-fully mature..." – add some specimen numbers to the parentheses (e.g. MOR xxx, MOR xxx; J. Wilson pers. obs.).

Description, Parietal, fourth page: "The term "EPS" is for now retained for clear reference to structures at homologous locations" – should this be analogous instead of homologous?

Phylogenetic analysis, first paragraph: “we added two characters to account for the degree of elongate” – elongate should be elongation, followed by a colon instead of a comma.

Taxon name: Beautiful.

Styracosaurus ovatus rediagnosis: “Monotypic, as for species” – does not apply here; you are trying to say that there are two species of Styracosaurus.

Discussion, Stratigraphy: Gould (2002) quote – include page number of quote.

Discussion, Stratigraphy: “an Achelousaurus-like pachyrhinosaur from the Lethbridge Coal Zone of Alberta (Ryan et al., 2010)” – Does this have a specimen number? “This specimen sits in the bottom” – Was it collected? Shouldn’t use present tense.

Discussion, Ontogeny, second sentence: Because Wilson et al 2015 is just an abstract, include the specimen numbers that provide this ontogenetic evidence.

Discussion, Geography, second page: “localities TM-023; TM-046...” – add MOR before “localities”.

Discussion, Taphonomy: Well described.

Discussion, Occurrences of anagenetic, second page: “Freedman Fowler” preferably has a space, not a hyphen.

Discussion, Diversity of Centrosaurine Ceratopsids, second sentence: “any given point in time maybe be” – change to “may be”.

Figure 1 caption: If the structures labeled EPS are not thought to be distinct epiossifications, then shouldn’t they be analogous rather than homologous?

Figures 1 & 2: Because two of the three bones in Figure 2 are the exact same photos as Figure 1, I suggest combining figures 1 and 2 to avoid duplication. Possibly Figure 3 could be combined into these as well, perhaps by using color or italics to differentiate the McDonald vs New numbers.

Figure 4: I understand the reasoning for having dorsal on the left and ventral on the right, so that the P# labels can be shared, but visually the figure would be much more intuitive if these were swapped left and right, because then the whole frill outline would be clear. The P# labels can be duplicated on each side. Also, the current font size of the P# labels seem huge. Looks like this is currently planned for full page width? One-column width might be enough, depending on whether just the shape outline matters, or whether you really need to see details of the bone texture.

Figure 6: I’d be interested in seeing an anterior view as well, to note the thickness (width) of the horncore.

Figures 9 & 10: Need a lot more information in the caption. Which matrix was used? Details of the methods? RI, CI? Which consensus tree is it, out of how many? I assume those are bootstrap numbers, but it needs explanation in caption. Also need to credit source of colored timeline, unless you made it completely from scratch.

Figure 9: Which point is the age of each taxon? Where the line ends? Where the name starts? What about uncertainties or ranges? You really don’t need the late Paleogene taking up space. Trim it, so you have room to stretch out the time periods that matter. The time scale doesn’t have

to be perfectly linear the whole way – certain areas can be compressed or stretched, as long as it is clearly marked.

Figure 11: What is on the y-axis? Time? Are the stratigraphic placements of taxa relative or absolute? Are the specimen placements based on Fowler 2017 or other data? Where does *ovatus* fit in? You mention in the methods that the exact strat is unknown, but you do know that it's from the Landslide Butte area, which narrows the range- is it enough to plot on this figure? If *ovatus* can't be plotted, say so in the caption.

Figure 12: Excellent. I really like the clear way this shows all of the missing evidence for cladogenesis, and the simple, parsimonious anagenesis hypothesis. This figure is a great “visual abstract”.

Reviewer: 2

Comments to the Author(s)

This more comprehensive approach to a species description is necessary in light of the complex (and largely unknown) challenges that ontogenetic changes in dinosaurs, particularly the ceratopsians, present to naming new species. The in-depth discussions on ontogeny, evolutionary trends, and stratigraphy are much appreciated in this case to supporting a new species. I wonder if, when considering the small sample sizes we see with many of these ceratopsians, anatomical variation among individuals can be significant enough to appear to represent more species than are actually present? Maybe someday we'll find out!

The species epithet is a very nice gesture and Carrie certainly deserves it - good thinking there.

Author's Response to Decision Letter for (RSOS-200284.R0)

See Appendix A.

Decision letter (RSOS-200284.R1)

06-Apr-2020

Dear Mr Wilson,

It is a pleasure to accept your manuscript entitled "A new transitional centrosaurine ceratopsid from the Upper Cretaceous Two Medicine Formation, Montana and the evolution of the 'styracosaur' dinosaurs" in its current form for publication in Royal Society Open Science.

Due to rapid publication and an extremely tight schedule, if comments are not received, your paper may experience a delay in publication. Royal Society Open Science operates under a continuous publication model. Your article will be published straight into the next open issue and

this will be the final version of the paper. As such, it can be cited immediately by other researchers. As the issue version of your paper will be the only version to be published I would advise you to check your proofs thoroughly as changes cannot be made once the paper is published.

on behalf of Professor Marcelo Sanchez (Associate Editor)
openscience@royalsociety.org

Appendix A

RSOS-200284 Response to referees

Reviewer suggestions are in normal text, author responses are highlighted.

Reviewer 1:

Ethics statement: These fossils came from someone's land, and either needed permits or permissions that should be mentioned here.

We have added an ethics statement at the end of the manuscript, between the end of the discussion and the author contributions section.

Abstract, lines 14-16: "currently considered valid" – Be careful with the phrasing here. Holmes et al 2020 refer *R. ovatus* to *Styracosaurus*. Of course you discuss this later in the paper, but in the abstract please use wording that everyone can agree on. Perhaps "herein considered valid", or "previously considered valid"?

We have modified this to read "previously considered valid" as suggested by the reviewer.

Introduction, paragraph 1, line 22: add "larger nasal horns" to the centrosaur list to balance the chasmosaur list.

This suggestion has been added.

Introduction, paragraph 2: "currently recognized" – Same phrasing issue with *Rubeosaurus* as in the abstract.

This has been adjusted in accordance with the previous suggestion.

Introduction: Excellent explanation of Unified Frames of Reference.

Thanks!

Institutional abbreviations: D.C. is missing a period.

A period has been added.

Materials and Methods: Phylogeny paragraph is too long and should be separated into smaller paragraphs. Also, it includes some results (posterior probabilities) that should be moved out of here. PAUP should have a citation: Swofford, D. L. 2003. PAUP*. Phylogenetic Analysis Using Parsimony (*and Other Methods). Version 4.0b10. Sinauer Associates, Sunderland, Massachusetts. MrBayes should have a source citation as well.

This paragraph has been divided into two shorter paragraphs. The sentence regarding posterior probabilities has been moved to the phylogeny section in the results. The correct PAUP citation has been added. The citation for MrBayes has been added.

Description, Parietal: I agree that this is a left lateral bar, not a right lateral bar.

Excellent!

Description, Parietal, second paragraph: “This morphology is seen in all other non-fully mature...” – add some specimen numbers to the parentheses (e.g. MOR xxx, MOR xxx; J. Wilson pers. obs.).

Specimen numbers have been added here.

Description, Parietal, fourth page: “The term “EPS” is for now retained for clear reference to structures at homologous locations” – should this be analogous instead of homologous?

This has been changed to “analogous”.

Phylogenetic analysis, first paragraph: “we added two characters to account for the degree of elongate” – elongate should be elongation, followed by a colon instead of a comma.

These changes have been made, though with semicolons, which is probably what the reviewer meant instead of colons.

Taxon name: Beautiful.

Thanks!

Styracosaurus ovatus rediagnosis: “Monotypic, as for species” – does not apply here; you are trying to say that there are two species of Styracosaurus.

This has been updated to just reflect the diagnosis of the species *S. ovatus*.

Discussion, Stratigraphy: Gould (2002) quote – include page number of quote.

The page number has been added.

Discussion, Stratigraphy: “an Achelousaurus-like pachyrhinosaur from the Lethbridge Coal Zone of Alberta (Ryan et al., 2010)” – Does this have a specimen number? “This specimen sits in the bottom” – Was it collected? Shouldn’t use present tense.

The specimen number has been added and the wording changed to reflect that it was collected, and in past tense.

Discussion, Ontogeny, second sentence: Because Wilson et al 2015 is just an abstract, include the specimen numbers that provide this ontogenetic evidence.

Specimen numbers have been added here.

Discussion, Geography, second page: “localities TM-023; TM-046...” – add MOR before “localities”.

MOR has been added here.

Discussion, Taphonomy: Well described.

Thanks!

Discussion, Occurrences of anagenetic, second page: “Freedman Fowler” preferably has a space, not a hyphen.

This has been changed to a space.

Discussion, Diversity of Centrosaurine Ceratopsids, second sentence: “any given point in time maybe be” – change to “may be”.

This change has been made.

Figure 1 caption: If the structures labeled EPS are not thought to be distinct epiossifications, then shouldn't they be analogous rather than homologous?

This change has been made.

Figures 1 & 2: Because two of the three bones in Figure 2 are the exact same photos as Figure 1, I suggest combining figures 1 and 2 to avoid duplication. Possibly Figure 3 could be combined into these as well, perhaps by using color or italics to differentiate the McDonald vs New numbers.

These figures were purposely created separately to help visually accompany the section in which we challenge the validity of “Rubeosaurus”. Because there are quite a few nuances to our arguments in that section, we wanted to very explicitly visually demonstrate each argument we make along that logical progression so that there is no confusion. That section progresses along a particular logical path, and the separate figures are intended to accompany each separate, though related, argument we make to demonstrate that MOR 492 is not referable to “Rubeosaurus”. We feel that blending these figures into one or two larger, more complex figures would not illustrate our arguments as clearly, and may be visually distracting from each individual point we are trying to make. We also think it is easier to direct readers to separate figures as we lay out our arguments rather than direct them to individual aspects of one large, more complex figure. For this reason, we would contend that for the sake of clarity, these figures remain separate so that readers can be directed to sequential figures as we lay out our findings.

Figure 4: I understand the reasoning for having dorsal on the left and ventral on the right, so that the P# labels can be shared, but visually the figure would be much more intuitive if these were swapped left and right, because then the whole frill outline would be clear. The P# labels can be duplicated on each side. Also, the current font size of the P# labels seem huge. Looks like this is

currently planned for full page width? One-column width might be enough, depending on whether just the shape outline matters, or whether you really need to see details of the bone texture.

We understand the reviewer's suggestion here and appreciate the idea of having the overall parietal shape replicated by switching the dorsal and ventral views. We tested this alternate arrangement, and it does partially show the overall parietal shape, but creates quite a bit of wasted white space in the top corners and in the middle between the photos. Without shrinking down the photos and moving them even further apart, the overall shape of the parietal is still only partially visible in the alternate arrangement, given that the specimen doesn't have the midline of the posterior parietal bar. Because RSOS publishes high resolution color figures, we wanted to maximize the size and resolution of the photos, which would in part be diminished by having to shrink the photos down and move them apart. We would like the figure to be full page width, as we would like as much detail visible, such as the vessel traces on the ventral surface and the differential dorsal vs ventral surface texture. Additionally, having the P3 processes in the middle under the suggested new arrangement leaves little room for the parietal line drawing, which has to be shrunk down considerably to fit into the reduced space in the middle.

Figure 6: I'd be interested in seeing an anterior view as well, to note the thickness (width) of the horncore.

Anterior view has been added.

Figures 9 & 10: Need a lot more information in the caption. Which matrix was used? Details of the methods? RI, CI? Which consensus tree is it, out of how many? I assume those are bootstrap numbers, but it needs explanation in caption. Also need to credit source of colored timeline, unless you made it completely from scratch.

This additional information on the matrix and methods used, as well as results, has been added to both figure captions.

Figure 9: Which point is the age of each taxon? Where the line ends? Where the name starts? What about uncertainties or ranges? You really don't need the late Paleogene taking up space. Trim it, so you have room to stretch out the time periods that matter. The time scale doesn't have to be perfectly linear the whole way – certain areas can be compressed or stretched, as long as it is clearly marked.

This information has been added stating that the taxon ages occur at the end of the branches, as well as a citation for source data for the taxon ages. The figure has additionally been trimmed to remove the unnecessary Paleogene space on the right.

Figure 11: What is on the y-axis? Time? Are the stratigraphic placements of taxa relative or absolute? Are the specimen placements based on Fowler 2017 or other data? Where does ovatus fit in? You mention in the methods that the exact strat is unknown, but you do know that it's from the Landslide Butte area, which narrows the range- is it enough to plot on this figure? If ovatus can't be plotted, say so in the caption.

This information has been added and clarified in the figure caption. *S. ovatus* is indeed omitted because its unknown stratigraphic placement prevents its stratigraphic relationship to *Stellasaurus*, *Einiusaurus*, and *Achelousaurus* from being plotted.

Figure 12: Excellent. I really like the clear way this shows all of the missing evidence for cladogenesis, and the simple, parsimonious anagenesis hypothesis. This figure is a great “visual abstract”.

Thanks!

Reviewer 2:

This more comprehensive approach to a species description is necessary in light of the complex (and largely unknown) challenges that ontogenetic changes in dinosaurs, particularly the ceratopsians, present to naming new species. The in-depth discussions on ontogeny, evolutionary trends, and stratigraphy are much appreciated in this case to supporting a new species. I wonder if, when considering the small sample sizes we see with many of these ceratopsians, anatomical variation among individuals can be significant enough to appear to represent more species than are actually present? Maybe someday we'll find out!

The species epithet is a very nice gesture and Carrie certainly deserves it - good thinking there.

Thank you! No revisions needed.